# Lessons learned from an fMRI-guided rTMS study on performance in a numerical Stroop task

Lysianne Beynel[1]* , Hannah Gura[1,2] , Zeynab Rezaee[1], Ekaete C. Ekpo[1], Zhi-De Deng[1], Janet O. Joseph[1,3], Paul Taylor[4], Bruce Luber[1], Sarah H. Lisanby[1]

1 Noninvasive Neuromodulation Unit, Experimental Therapeutics Branch, Intramural Research Program, National Institute of Mental Health, Bethesda, Maryland, United States of America, 2 Neuroscience Graduate Group, Perelman School of Medicine, University of Pennsylvania, Philadelphia, Pennsylvania, United States Of America, 3 Pathobiology Graduate Program, Division of Biology and Medicine, Brown University, Providence, Rhode Island, United States of America, 4 Scientific and Statistical Computing Core, Intramural Research Program, National Institute of Mental Health, Bethesda, Maryland, United States of America

☯ These authors contributed equally to this work.
* Lysianne.beynel@nih.gov

**Data Availability Statement:** Behavioral data are available from the Figshare database: https://

## Abstract

The Stroop task is a well-established tool to investigate the influence of competing visual categories on decision making. Neuroimaging as well as rTMS studies have demonstrated the involvement of parietal structures, particularly the intraparietal sulcus (IPS), in this task. Given its reliability, the numerical Stroop task was used to compare the effects of different TMS targeting approaches by Sack and colleagues (Sack AT 2009), who elegantly demonstrated the superiority of individualized fMRI targeting. We performed the present study to test whether fMRI-guided rTMS effects on numerical Stroop task performance could still be observed while using more advanced techniques that have emerged in the last decade (e.g., electrical sham, robotic coil holder system, etc.). To do so we used a traditional reaction time analysis and we performed, post-hoc, a more advanced comprehensive drift diffusion modeling approach. Fifteen participants performed the numerical Stroop task while active or sham 10 Hz rTMS was applied over the region of the right intraparietal sulcus (IPS) showing the strongest functional activation in the Incongruent > Congruent contrast. This target was determined based on individualized fMRI data collected during a separate session. Contrary to our assumption, the classical reaction time analysis did not show any superiority of active rTMS over sham, probably due to confounds such as potential cumulative rTMS effects, and the effect of practice. However, the modeling approach revealed a robust effect of rTMS on the drift rate variable, suggesting differential processing of congruent and incongruent properties in perceptual decision-making, and more generally, illustrating that more advanced computational analysis of performance can elucidate the effects of rTMS on the brain where simpler methods may not.

## Introduction

The Stroop task is used to investigate the effects of competing visual categories on decision making. The earliest version of the Stroop task [1] employed words printed in various colors,

figshare.com/articles/dataset/
NumericalStroopTMS_xlsx/25130669.

**Funding:** Beynel L., Gura, H., Rezaee, Z., Ekpo, E., Joseph, J., Deng, Z-D., Luber, B., and Lisanby, S.H. are supported by the NIMH Intramural Research Program (ZIAMH002955). Taylor, P. was supported by the NIMH Intramural Research Program (ZICMH002888) of the NIH/HHS, USA. The funders had no role in study design, data collection and analysis, decision to publish, or preparation of the manuscript.

**Competing interests:** "The opinions expressed in this article are the authors' and do not reflect the views of the National Institutes of Health, the Department of Health and Human Services, or the United States government. Dr. Deng is inventor on patents and patent applications on electrical and magnetic brain stimulation therapy systems held by the National Institutes of Health (NIH), Columbia University, and University of New Mexico. Dr. Lisanby is inventor on patents and patent applications on electrical and magnetic brain stimulation therapy systems held by the NIH and Columbia University, with no remuneration. Our competing interests do not alter our adherence to PLOS ONE policies on sharing data and materials."

and participants were asked to name the colors of the ink that the words were printed in as quickly as possible. When the ink was the same color as the word (the congruent case), naming responses were faster, while when the ink color was different (the incongruent case), responses were slower. These facilitatory and interference effects in performance are quite reliable in their general replicability across healthy adults and in their stability in effect size (after a short initial learning period) within individuals. This reliability has made the Stroop task a useful tool for investigating the cognitive processes involved with its effects. For example, automaticity in stimulus recognition, processing speed, and selective attention, as well as in neuropsychological testing in patients with deficits in such processing. The reliability of this task has also enabled neuroimaging investigations into the neural structures underlying Stroop effects, which have implicated the roles of the parietal cortex and prefrontal regions such as the anterior cingulate cortex [2, 3]. Studies employing online repetitive transcranial magnetic stimulation (rTMS) during task performance have sought to establish causal relationships between these neural structures thought to subserve Stroop effects.

In one such study, a counting Stroop task was used, in which subjects were asked to count the number of words shown on a screen, comparing screens using number words (the interference condition) and screens using neutral words [4]. Short (4 pulse) trains of rTMS given at 10 Hz starting 200 ms after stimulus onset to the anterior cingulate cortex caused the eradication of the slowing effect of the interfering words compared to the neutral words, while significant slowing was shown with stimulation to an active control site in posterior midline cortex. Another set of studies used a numerical Stroop task [5, 6] in which two digits were concurrently shown, and the subject was to identify the larger number in magnitude (i.e., 4>2). To induce a Stroop effect, the physical size of each number was manipulated. In the neutral condition, both numbers had the same font size. However, in the congruent condition, the number larger in magnitude had the larger font size; and in the incongruent condition the number smaller in magnitude was in larger font. Both studies (performed by the same group), applied a short rTMS train (3 pulses) at 10 Hz, beginning 220 ms after stimulus onset over the right intraparietal sulcus (IPS). Both resulted in significant reduction of the Stroop facilitation/interference effects on reaction time compared to sham stimulation. Overall, there was a consistency across these three studies using rTMS to modify performance in Stroop tasks: in their application of online rTMS, their effects on Stroop performance, and in the interpretation of their results. In terms of the latter, in Cohen Kadosh et al. [6] it was suggested that rTMS reduced the Stroop interference effect by disrupting right parietal automated processing associated with the competing but irrelevant stimulus dimension which would normally slow the decision process involved in completing the task. This suggestion could also apply to Sack et al. [5] as well as Hayward et al. [4] all of which could be classified under a general mechanism of performance enhancement caused by "addition by subtraction", where a normally competing but interfering process is disrupted by rTMS [7].

The primary goal of the Sack et al. [5] study was to provide a direct comparison of four different rTMS targeting approaches used at that time: 1) scalp-based measurement, 2) anatomical MRI locations, 3) average group coordinates from fMRI activations using the task being studied, and 4) individualized targeting based on task-related fMRI. The interindividual reliability of the numerical Stroop task they successfully used in Cohen Kadosh et al. [6] made it a good candidate task for the targeting comparison. Sack et al. [5] concluded that the most effective approach was individualized targeting based on the task-related fMRI. This was an important result for the brain stimulation field in general, as one of the main moderators of rTMS efficacy is the targeting approach [8]. More than a decade after this elegant demonstration of the importance of individualized fMRI targeting, experimental TMS methodologies have advanced, with for example the development of robotic coil holders that allow for more precise

and stable positioning of the coil relative to the stimulation target, and an electrical sham technique that mimics the TMS-induced sensations, and thus potentially better blinds subjects to experimental conditions. In the present study, we tested whether the addition of these techniques combined with individual fMRI-based targeting approach endorsed by Sack and colleagues could improve rTMS effect on numerical Stroop performance, with the expectation based on Cohen Kadosh et al. [6] and Sack et al. [5] that rTMS applied over the right IPS would decrease the interference of incongruent stimuli and thus lessen of the reaction time in that condition. Additionally, contrary to the prior study from Sack et al., participants completed the task multiple times preceding rTMS application (one block of trials before the MRI session, 4 blocks during MRI, and one block in the TMS session prior to TMS), to stabilize performance and minimize variability due to practice effects. This approach enabled us to probe the influence of task practice on behavioral performance and fMRI activations.

Finally, we decided post-hoc to perform behavioral modeling approach using a drift diffusion model to assess rTMS effects within a decision-making context. This model, by considering the distribution of reaction times rather than just central tendencies, offers more insight in decision making processes. By elucidating the dynamics of evidence accumulation over time, and by distinguishing between decision and non-decision processes, the model provides a more nuanced understanding of ongoing cognitive mechanisms. Drift diffusion models have demonstrated effectiveness in modeling performance data derived from simple tasks with short reaction times ($< 1$ s), as in the present case. In addition, they have been linked to neural behavior on a number of scales, from single cells to large populations (e.g., using EEG and fMRI) [9]. The drift diffusion models have also been successfully used to investigate the neuronal substrates for numerical representation using TMS [10]. Therefore, it was expected that this modeling approach would reveal more subtle rTMS effects, as recommended by Hartwigsen et al. [11].

## Material and methods

### Participants

Nineteen healthy volunteers (14 female, 5 male) with a mean age of 39 ± 14 (SD) years participated in this 2-day study pre-registered on CT.gov (NCT00024635). All participants were right-handed and identified as White (n = 10), Black (n = 4), Asian (n = 3), or Hispanic (n = 2). Participants were recruited from the Washington, DC, metropolitan area via flyers, listserv emails, advertisements, and the NIH Office of Patient Recruitment. All participants gave written informed consent approved by the NIMH Institutional Review Board (#18-M-0015) to be screened with psychiatric, physical, and neurological examinations, urine drug screens, and pregnancy tests for women of childbearing capacity. Volunteers were excluded if they had a history of current or past Axis I psychiatric disorders (including substance abuse/dependence) as determined by the Structured Clinical Interview for DSM-IV Axis I Disorders (SCID-NP), a history of neurological disease, or seizure risk factors. Participants were compensated for their participation.

### Experimental design and numerical Stroop task

Participants came in for two experimental sessions. The first visit included consenting, an initial numerical Stroop task practice session, and structural and functional neuroimaging in combination with the Stroop task. The second visit included an additional Stroop task practice followed by active and sham rTMS, with the order counterbalanced between participants (Fig 1). The numerical Stroop task was coded using the PsychoPy Builder interface [12]. In this task, two single-digit numbers were presented on the screen (between 1 and 8 numbers apart)

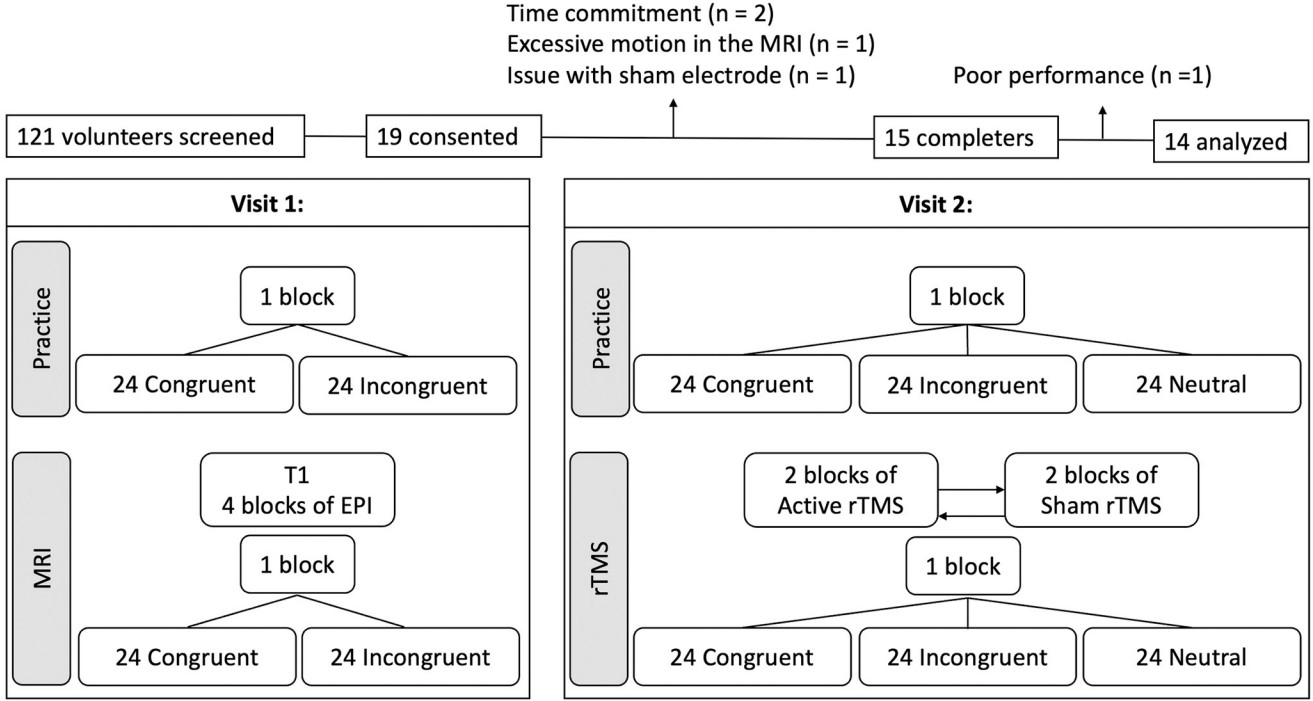

**Fig 1. Consort diagram and experimental design.**

with a center-to-center horizontal visual angle of 7.7 degrees. The numbers varied in physical size (vertical visual angle of 1.4 degrees for the small numbers or 2.9 for the larger ones). Three conditions were tested: Congruent, in which the numerically larger digit was also larger in size; Incongruent, in which the numerically smaller digit was larger in size; Neutral, in which both digits presented were the same small size. Participants were asked to indicate with a left or right button press as quickly and accurately as possible the number with the numerically larger magnitude, irrespective of size. Reaction time and accuracy were collected.

**Visit 1.** After consenting to the study, participants completed a first training block of the numerical Stroop task with 48 trials (24 congruent, 24 incongruent; no neutral trials were included as they were not used during the fMRI acquisition). Feedback was provided after each trial with the word "correct" or "oops" on the screen to ensure participants understood the task, and overall accuracy was provided on the screen at the end of the task. Participants were then scanned using a 3T gradient echo scanner (General Electrics) equipped with a 32-channel head coil. A structural T1-weighted MP-RAGE image was first acquired (FOV = 25.6 cm$^2$, voxel dimension = 1 mm isotropic, TE = Min Full echo, TI =1100 ms, bandwidth = 25 Hz/Pixel, flip angel = 7 degrees). Four EPI sequences were then acquired with an oblique orientation defined with the AC-PC axis (FOV = 72*72*52, voxel dimension = 3 mm isotropic, TE = 30ms, TR = 3000 ms, flip angle = 70 degrees) during which participants performed four more blocks of the Stroop task. Feedback (accuracy and average reaction time) was provided at the end of each block. Stimuli were back-projected onto a screen located at the foot of the MRI bed using an LCD projector. Subjects viewed the screen via a mirror system located in the head coil and the start of each run was electronically synchronized with the MRI acquisition computer. BOLD signal analyses were then performed (see MRI targeting section) to identify a target for TMS.

**MRI targeting.**   AFNI (version 22.3.05) [13] was used to process MRI data. First, @SSwarper was used for skull stripping and estimating nonlinear alignment (via 3dQwarp; [14]) of the anatomical image from native space to Montreal Neurological Institute (MNI) stereotaxic space [15]. Then afni_proc.py was used to setup a full pipeline for the fMRI analysis of each participant, including the automatic generation of a quality control HTML for evaluating the data and processing steps [16]. The full afni_proc.py command is provided in GITHUB (https://github.com/afni/apaper_rtms_fmri_stroop) but we briefly describe the steps here. The first 2 TRs (6 seconds) to allow stabilization of the magnetic field were discarded from the analysis. All images were corrected for slice acquisition timing, motion corrected by registration to the minimum outlier volume, and spatially smoothed with a 4mm full-width-half-maximum smoothing kernel. EPI-anatomical alignment was performed using the lpc+ZZ cost function [17] with local EPI unifizing for additional stability, and these datasets were checked for left-right consistency [18]. To reduce effects of participant motion, volumes with large motion (Enorm > 0.3 mm between successive time points) were censored.

A duration modulation regression basis was used, and separate events were modeled for Congruent and Incongruent trials. The onset for each was defined by the beginning of each event, and duration for each was defined by the extent of the participant's reaction times. Incorrect trials in which participants did not answer correctly (for either Congruent or Incongruent stimuli) were modeled identically but as a separate stimulus class, and missed trials were not modeled. The contrast between Incongruent > Congruent trial was generated. After regression modeling, several steps were then performed for QC evaluation using the APQC HTML, such as: checking the alignment between the anatomical, EPI and the template image; checking that the stimuli were properly assigned between each stimulus class; checking how many data points were censored because of motion. One participant was excluded via this QC evaluation because of motion (with 27.5% of the data censored due to motion); looking at stimulus statistical maps for validity. The final Incongruent > Congruent statistical maps and the intraparietal sulcus (IPS) mask from Neurosynth (https://neurosynth.org/) were then mapped back into subject original space; and along with the original T1 these three results were loaded into the neuronavigation system (BrainSight, Rogue Research, Canada); and the area of the statistical peak within the right IPS mask for each participant was chosen as the rTMS target.

**Visit 2.**   During visit 2, participants conducted a training block followed by four blocks of the numerical Stroop task in conjunction with active or sham rTMS. Active and sham stimulation were applied on the same day in random order, counterbalanced between participants (e.g., 2 blocks of active followed by 2 blocks of sham or vice-versa). Training and stimulation blocks contained 72 trials (24 congruent, 24 incongruent, and 24 neutral trials, replicating Sack's number of stimuli), with feedback (accuracy and average reaction time) provided at the end of the block.

Stimulation was applied at 10-Hz over the individualized IPS target at 60% maximum stimulator output (MSO) as per Sack et al [5]. During each trial, a triplet of pulses was applied at 220, 320, and 420 ms after the onset of numbers presentation. Sham stimulation was applied using the same coil in placebo mode, which produced clicking sounds and somatosensory sensations via electrical stimulation with scalp electrodes similar to the active mode, but without a significant electric-field (E-field) induced in the brain [19]. Each subject was informed that 2 types of stimulation would be applied, and that we needed to define the intensity. For active rTMS, resting motor threshold (rMT) was assessed using motor threshold assessment software tool MTAT [20] over the right motor cortex and used as a potential covariate. For sham stimulation, subjects were asked if they felt each pulse. If not, the sham intensity was increased. If the pulse was felt, participants were asked if the sensation was tolerable. The lowest tolerable intensity was chosen for stimulation. An A/P-B65 coil powered by a MagVenture MagPro

X100 stimulator (MagVenture, Farum Denmark) was used, and coil placement was guided by a computerized frameless stereotaxic system (Brainsight, Rogue Research, Montreal Canada). Coil position and orientation was maintained precisely over the target with the Axilum Robotics TMS-Cobot (Axilum Robotics, Schiltigheim, France).

**Statistical analysis.** The analysis is divided in three sections: the first part relies on classical analysis of reaction time and accuracy; the second section uses a drift diffusion modeling approach to test the rTMS effect with a more subtle decision-making context. The final section is exploratory and seeks to identify potential predictors of the rTMS effect. Analyses of accuracy and reaction time of correct trials were performed with JASP (Version 0.17.2, Amsterdam Netherlands). In the classical analysis, three analyses were performed: the first linear mixed model investigated the impact of task practice on behavioral performance using Congruency (congruent, incongruent) and Timepoint (1 block before MRI, 4 separate MRI Blocks, 1 block before TMS) as fixed effects variables, and Participants as a random effect variable. The second analysis used a linear mixed model to investigate the rTMS effect on task performance using participants as a random effect and Stimulation (active, sham), Congruency (congruent, incongruent, neutral) and Order of stimulation (sham-first, active-first) as fixed effects. Information about the pre-processing of behavioral data is provided in the results section. The third analysis was conducted to replicate Sack et al.'s finding on the size congruency effect (SCE), the difference between reaction time for Incongruent and Congruent trials, to test whether active rTMS would decrease the SCE compared to sham stimulation. We also investigated whether rTMS impacted the facilitatory and inhibitory component of the SCE. For each of these analyses, we performed an ANOVA with Stimulation (active, sham) as a within-subject factor, and Order of stimulation (active first, or sham first) as a between-subject factor.

While changes in reaction time and/or accuracy are the most commonly used variables to investigate rTMS effects on a behavioral task, it has been demonstrated that they only provide a limited statistical sensitivity, which might not pick up slight changes in response strategies, for instance conflicting response tendencies in Stroop tasks (see [11] for a review). To overcome this limitation, the second part of the analysis uses a drift diffusion model [21], in which the reaction time distributions for correct and incorrect responses were fit into a model comprised of the sensitivity to the relevant stimulus (drift rate), the decision threshold (boundary parameter, i.e., the amount of information needed to trigger the button press), and the non-decision time (i.e., the duration of information processing before the decision process and the time taken to execute the motor command). This approach allows assessment of rTMS effects within a more subtle decision-making context. Drift diffusion models were fit to trial-by-trial measures of reaction times and response type. The drift rate, boundary separation, and nondecision time parameters were estimated separately across congruency and TMS conditions. 5% of all trials were assumed to be outliers and modeled under a different process.

Finally, a linear regression model was performed to explore the impact of participant's age, stimulation intensity, distance to scalp, fMRI activation at the stimulated target location, distance to Sack et al.'s group fMRI target, and number of days between the MRI and the TMS session, as potential predictors of rTMS effect, and to test the potential impact of numerical distance between stimuli, a factor that is known to strongly impact reaction time.

## Results

Four subjects were withdrawn during the protocol due to the time commitment (n = 2), movement during the MRI (n = 1), and hair thickness that was incompatible with accurate electrode placement (n = 1). Data from one additional subject was excluded due to excessively long reaction times, which were greater than 2 standard deviations above the group mean for four out

of six conditions. Therefore, analyses were performed on 14 participants. All data below are presented as mean ± standard error.

## Classical analysis

**Practice effect.** To investigate the impact of practice on task performance, we applied a linear mixed effect model to analyze correct reaction time (Fig 2A). Before performing the analyses, outliers and missed trials were removed from the data. First, missed trials, in which participants did not respond, were removed from both for reaction time and accuracy data (4.2% of the trials). Next, for the analysis of reaction time only, incorrect trials, in which participants answered incorrectly were removed (1.9%); then trials with reaction times larger than 2.5 standard deviations above the mean were removed (2.3%). We did not run any model on the accuracy data, as there was not enough variability in the data because accuracy was near ceiling at all timepoint (overall group accuracy was 98.1 ± 0.3%, see Fig 2B for accuracy values and change across sessions). Results below focus on correct reaction time.

We assessed the effects of Congruency and Timepoint on correct reaction times during practice using a linear mixed model with Congruency and Timepoint as fixed effect variables, and Participant as random effect. The linear mixed model did not show any interaction between Congruency and Timepoint, however we found the expected main effect of Congruency ($F_{(1,12.58)} = 94.698$, $p < .001$) with congruent trials yielding faster reaction times (533 ± 16 ms) than incongruent trials (607 ms ± 14 ms). A main effect of Timepoint ($F_{(5, 14.68)} = 23.830$, $p < 0.001$) was also observed; and post-hoc contrasts (Bonferroni corrected) revealed that while performance in the pre-MRI practice block was not significantly different from the first MRI block ($p > 0.05$); reaction times at each of the timepoints thereafter were faster than the initial reaction time collected before the MRI ($p < 0.05$ for all pair-wise comparisons). Reaction times at all timepoints after MRI block 1 were also significantly faster than for MRI block 1 ($p < 0.05$ for all pair-wise comparisons). No significant pairwise differences in RT between other blocks were observed (Fig 2A).

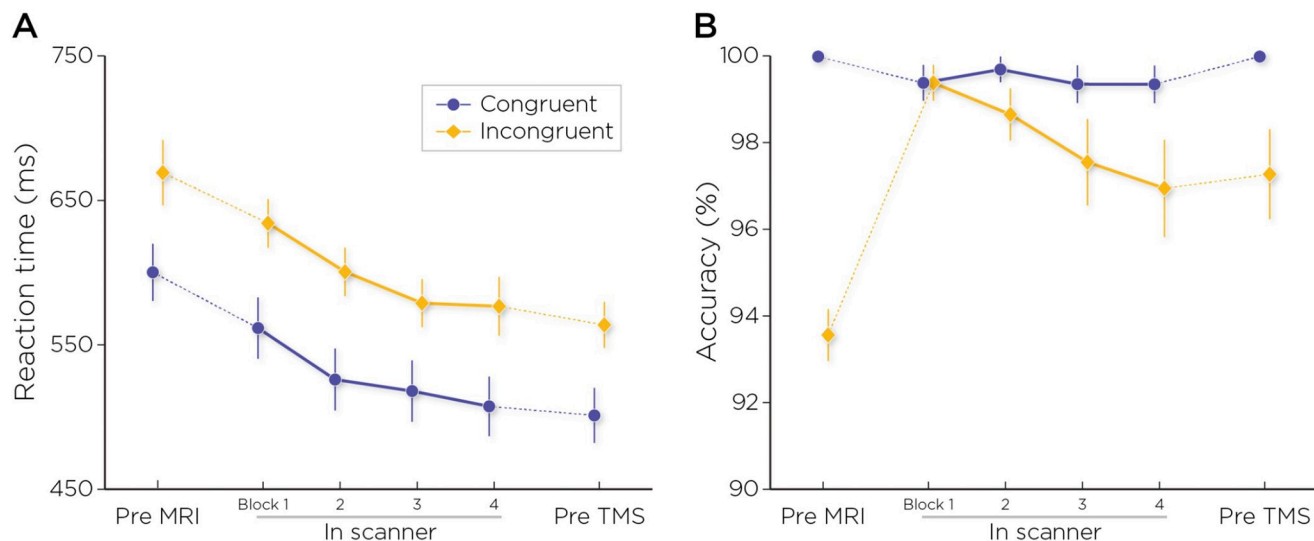

**Fig 2. Behavioral performance in the scanner: A) reaction times and B) accuracy in the six training blocks, for congruent trials in blue and incongruent trials in yellow.**

In addition, as our group of participants was older (39 years old on average) than the participants in Sack's study (24 years old on average), we investigated whether there was any correlation between age and reaction times in Congruent and Incongruent trials collapsed across the six practice blocks. Results did not reveal any significant correlations between them (Congruent trials: r = 0.26; p = 0.40; Incongruent trials: r = 0.45, p = 0.13), suggesting that age did not impact performance.

**rTMS effect.** The same cleaning approach was performed on the rTMS data except that an additional first step was added—the trials in which the TMS coil was far away from target (>= 3mm, often due to participant movement) were removed from the data (2.1% of trials). Then, trials in which participants did not answer were excluded (1.3% of remaining trials). Finally, for reaction time analysis, trials in which participants did not answer correctly (12%) and trials with reaction times larger than 2.5 standard deviations above the mean (3% of remaining trials) were removed. The overall group accuracy was 88%, however since no main effects or interaction were found between the three variables, the results below focus on reaction times only. A linear mixed model was performed with participants as a random effect, and Stimulation, Congruency and Order as fixed effects. The model revealed a main effect of Congruency (F(2, 25.30) = 47.56, p < 0.001), with incongruent trials slowing down participants (492 ± 15 ms) compared to congruent (439 ± 12 ms) and neutral trials (457 ms ± 12 ms) (p < 0.01 for each pair-wise comparison). The model also revealed a significant interaction with Stimulation, Congruency and Order (F(2, 44.43) = 3.68; p = 0.03). However, decomposition of this interaction with post-hoc Bonferroni-corrected contrasts did not reveal any significant difference between active and sham rTMS.

**Size congruency effect (SCE).** To replicate the findings of Sack et al. [5], we calculated the SCE by measuring the reaction time (RT) difference between congruent and incongruent trials [SCE = RT (incongruent)—RT (congruent)]. We excluded data from an additional participant whose size congruency values exceeded 2.5 standard deviations above the group mean, analyses were therefore performed on 13 participants. Contrary to expectation, the repeated measures ANOVA did not reveal a main effect of Stimulation (F(1,11) < 1, p = 0.40: Active = 51 ± 16 ms; Sham = 46 ± 9 ms), Order (F(1,11) < 1, p = 0.58) or interaction between Order and Stimulation (F(1,11) < 1, p = 0.58), suggesting that contrary to Sack et al., rTMS in our study did not impact the size congruency effect (Fig 3).

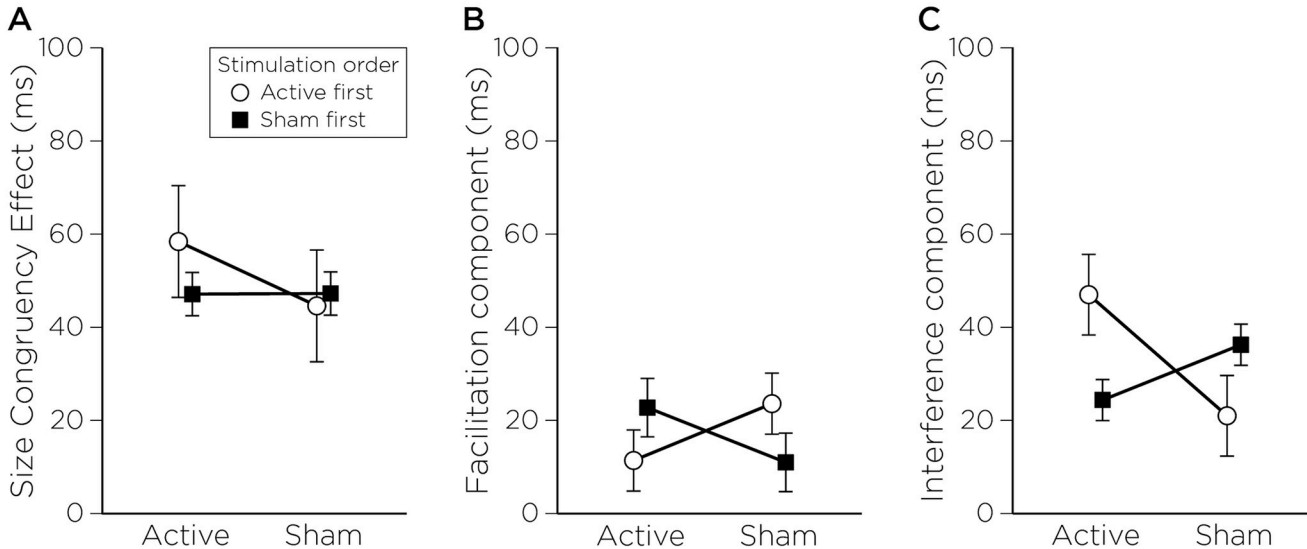

**Fig 3. rTMS effect of size congruency effect and its components.** A) Size Congruency effect, B) Facilitatory component and C) Interference component showing a significant interaction between Stimulation and Order of stimulation.

Following the methodology employed by Sack et al., we also calculated the facilitatory component as [RT (neutral)—RT (congruent)] and the interference component computed as [RT (incongruent)—RT (neutral)]. For the facilitatory component, neither the interaction between Stimulation and Order, nor the main effects reached statistical significance. However, for the interference component, a significant interaction between Stimulation and Order was observed (F(1, 11) = 9.39; p = 0.011, $\eta^2$ = 0.21). However, post hoc analysis did not reveal any significant differences among pairwise comparisons (Active-first group: Active = 47 ± 26 ms, Sham = 21 ± 16 ms; Sham-first group: Active = 24 ± 17 ms, Sham = 36 ± 17 ms). The absence of broader significant findings may be attributable to an insufficient sample size, which could diminish the power of our analysis.

## Computational analysis of rTMS effects with a drift diffusion model

The drift diffusion model revealed a significant interaction between Congruency and Stimulation (F(2,70)=10.9, p<.001, Fig 4). Post hoc comparisons revealed that while no difference were observed for incongruent (t(14)=0.90, p=.39) and neutral trials (t(14)=0.89, p=.39), the drift rate was significantly higher with active rTMS compared to sham stimulation for the congruent trials (t(14)=6.5, p<.001). While we would have expected rTMS to the right IPS to increase the drift rate for both congruent and incongruent trials, given its role in automatic spatial magnitude processing; the current finding suggests that within the decision process, at least in a well-practiced state, the mechanism underlying the integration of congruent properties differs and can be more easily distinguished from that underlying the integration of incongruent properties.

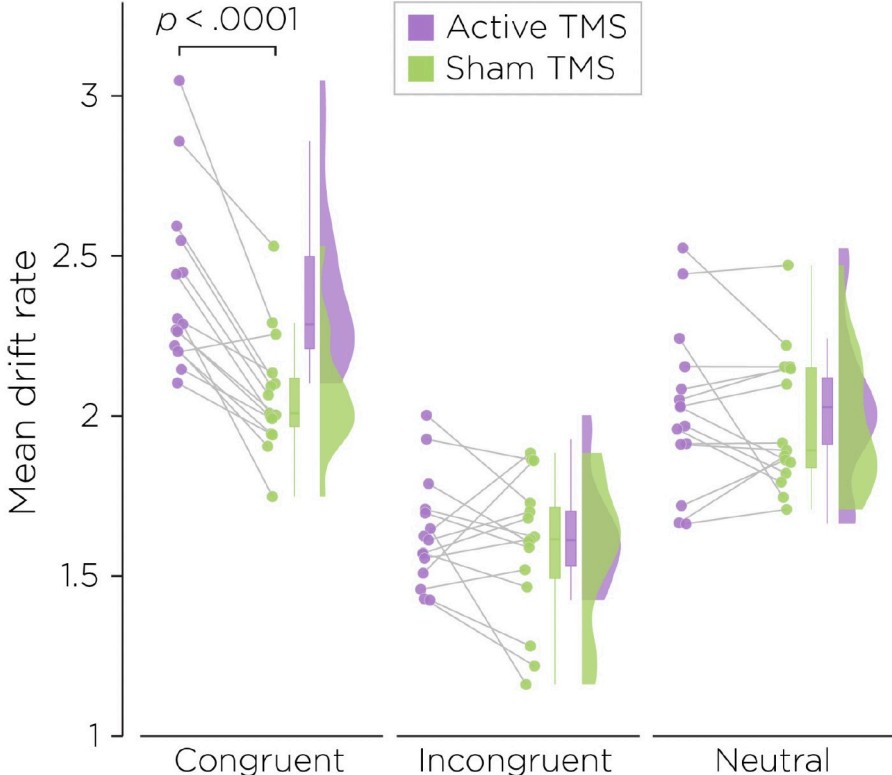

**Fig 4. Mean drift rate for each congruency and each stimulation condition.**

## Exploratory analyses

In the first analysis, we performed a linear regression analysis with a backward selection method to test the impact of several covariates that could impact rTMS effect (participant's age, functional activations at the stimulated target, scalp-to-cortex distance, distance to Sack's targets, stimulation intensity expressed as a percentage of the resting motor threshold, and number of days between MRI acquisition and TMS session) on rTMS effect for calculated as follow:

$$rTMSeffect(\%) = \frac{ReactionTimewithActiverTMS - ReactionTimewithShamrTMS}{ReactionTimewithShamrTMS} \times 100$$

This method allowed to iteratively remove non-significant variables from the model, and thus to identify the most relevant predictors of rTMS effect while controlling for potential confounding variables. The final model showed that only stimulation intensity predicted the rTMS effect (F(1,13) = 13.09, p = 0.004, adjusted $R^2$ = 48.2%). This suggests that participants who received the strongest stimulation intensity where the ones benefitting the most from rTMS (showing the fastest reaction times with active rTMS). We note that on average, in our sample, an intensity of 60% MSO led to a strong variability in stimulation intensity, which ranged from 94% to 188% rMT (mean = 128.9 ± 5.7% rMT, see S1 Table for individual participants data). None of the other variables: participant's age, scalp-to-cortex distance, distance to Sack's target, and number of days between fMRI acquisition and TMS session, or fMRI activation at the stimulated target predicted the rTMS effect. Contrary to our assumption analysis of the fMRI activation at the group level, did not show the expected right IPS activation. Instead, the analyses of BOLD signal revealed a cluster of activation (t > 2.10, with 23 voxels) in the left IPS and in subcortical structures such as the bilateral corpus callosum, and the left cingulum (see Fig 5A and S2 Table for the list of clusters and their overlap with the Glasser atlas, generated by the AFNI whereami function).

To evaluate whether the lack of rTMS effect may have been due to targeting issues resulting from a change in fMRI activation across blocks, which may have resulted from practice, learning, and/or fatigue, we investigated how activations differed between the first and last pairs of blocks. Our results showed that while activation in the bilateral IPS was strong during the first two blocks, IPS activity vanished during the last two blocks (Fig 6). Given that these changes were observed while accuracy remained quite high and constant across all blocks, and while reaction time continued to improve (Fig 2A). We suspect this change is due to a learning effect and hypothesize two potential mechanisms: 1) the IPS was highly involved in the earlier learning stages of the task and became less necessary with practice, with possible activity shifts to other regions; 2) the number of IPS neurons necessary for task performance decreased with the optimization caused by practice, with subsequent diminishment of signal. If correct, these hypotheses indicate the need for a better understanding of the local and network changes associated with task practice and learning effects to estimate the best targeting parameters. In the present case we potentially would have seen stronger rTMS effects if we only acquired two blocks of fMRI, and this may have served as a better target for rTMS if the latter of the two reasons stated above for the activation decrease was true.

The second analysis investigated the effect of numerical distance between stimuli. Indeed, it has been demonstrated in the literature that performance in the Stroop task is inversely correlated with numerical distance; more precisely, if the distance between two numbers is large (e.g. between 1 and 9: distance of 8) reaction times are faster than if the distance is small (e.g., between 2 and 3: distance of 1) [23]. Since our study was not designed to test this effect, the distribution of the trials in each distance prevented us from testing the effect of each distance on

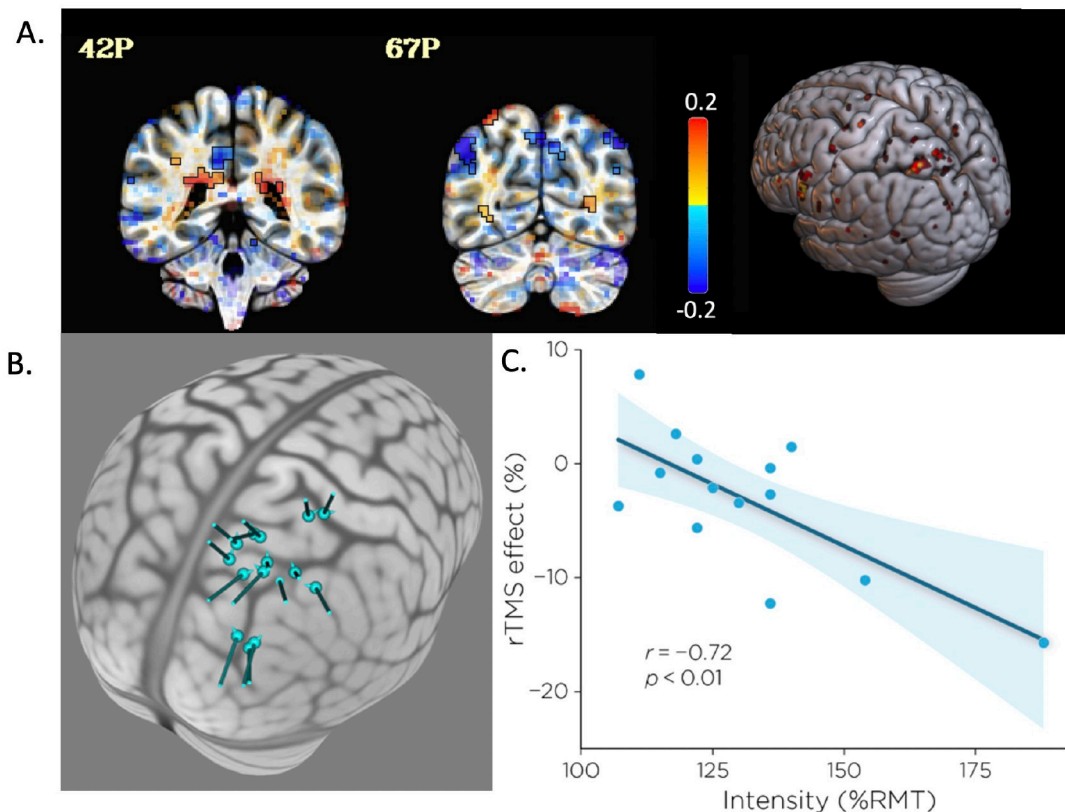

**Fig 5. fMRI results, TMS targets, and correlations.** A) Group fMRI activation in the Incongruent > Congruent contrast. Colors show the effect estimate value. Each voxel is thresholded by its t-stat values and the transparent thresholding was applied [22] (Threshold: t = 2.11, bi-sided; p < .05) on a coronal and an axial view and on a 3D rendering (left is subject left). B) TMS target for each individual on an MNI brain, and C) correlation between stimulation intensity and rTMS effect.

performance. Therefore, we decided to categorize trials into small distance (from 1-3 digits) and large distance (from 4-8 digits) groups and performed an ANOVA with Numerical Distance (Small and Large), Stimulation (Active and Sham). The analysis revealed the expected main effect of Numerical Distance F(1,13) = 51.83, p < 0.001) with faster reaction times for large (448 ± 12 ms) compared to small distances (471 ± 13 ms). The main effect of Simulation Type was not significant (F(1,13) = 3.46; p = 0.09), and interestingly no interaction was found between Stimulation Type and Numerical Distance (F(1,13) = 1.14; p = 0.31), suggesting that while Numerical Distance had a strong impact on reaction time, the effects of rTMS were not impacted by it.

## Discussion

Cohen Kadosh et al. [6] and Sack et al. [5] reported that using short bursts of 10 Hz rTMS to right IPS during the performance of a numerical Stroop task caused a significant decrease in the difference between incongruent and congruent RTs. We were not able to replicate these findings with a larger sample size, seeing no main effect of rTMS on RT or on the SCE. As discussed below, this failure to replicate may have to do with methodological differences between their studies and ours. However, we did observe a significant effect of right parietal rTMS on task performance, with thirteen of fourteen subjects showing an increase with active stimulation to the drift rate variable in a drift diffusion model of the RT data.

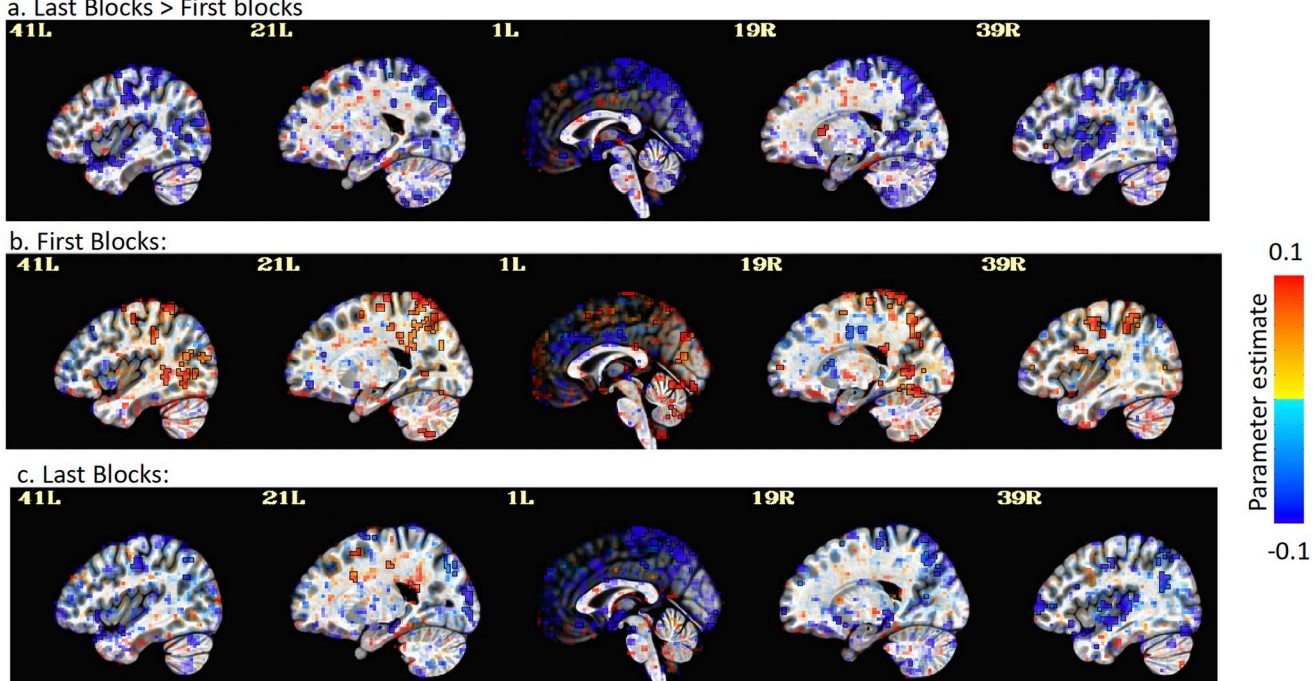

**Fig 6. fMRI activation in the Incongruent > Congruent contrast in the first pair and last pairs of fMRI acquisition.** Colors show the effect estimate value for: **a)** Last blocks > First blocks, **b)** the First blocks only, and **c)** the Last blocks only. Each voxel is thresholded by its t-stat values and the transparent thresholding was applied [22] (Threshold: t = 2.16, bi-sided; p < .05) on a sagittal view.

Use of more sophisticated approaches to modeling performance data stands a much greater chance of revealing effects of rTMS on behavior [11]. In the present case, rather than just looking at mean RT effects, use of a drift diffusion model incorporates the full RT distributions of both correct and incorrect responses, accommodating speed/accuracy trade-offs and bias effects, and by assuming that information accumulates at a constant rate, is able to estimate that rate [9]. This drift rate can be thought of as an estimate of the signal-to-noise ratio in the perceptual decision process.

Drift diffusion modeling has previously been used successfully with rTMS. For example, 1 Hz rTMS to dorsolateral prefrontal cortex, which might be expected to disrupt ongoing decision processing occurring there, was reported to decrease the drift rate found modeling data from the performance of an object discrimination task [24]. In the context of numerical Stroop task performance, rTMS to right IPL might be expected to increase the drift rate for both congruent and incongruent trials. This is based on the role of right parietal cortex in automatic spatial magnitude processing (in the case of the numerical Stroop task, a comparison of physical sizes) that Cohen-Kadosh et al. [6] posited to explain their TMS findings. Right IPL rTMS should disrupt the automatic processing of number font size, leading to the subtracting out of its irrelevant contribution to the ongoing decision process. The decision process, having one less channel of useless information (i.e., noise) to integrate, is thus made more efficient, which should be reflected in an increased drift rate. In this interpretation for the cause of a change in drift rate with rTMS, the congruency condition should not matter. However, here only the congruent trials, but not the incongruent, showed this effect. In the context of numerical Stroop task performance, rTMS to right IPL might be expected to increase the drift rate for

both congruent and incongruent trials. This is based on the role of right parietal cortex in automatic spatial magnitude processing (in the case of the numerical Stroop task, a comparison of physical sizes) that Cohen-Kadosh et al. [6] posited to explain their TMS findings. Right IPL rTMS should disrupt the automatic processing of number font size, leading to the subtracting out of its irrelevant contribution to the ongoing decision process. The decision process, having one less channel of useless information (i.e., noise) to integrate, is thus made more efficient, which should be reflected in an increased drift rate. In this interpretation for the cause of a change in drift rate with rTMS, the congruency condition should not matter. However, here only the congruent trials, but not the incongruent, showed this effect. A primary methodological difference between the Sack et al. [5] and Cohen-Kadosh et al. [6] studies and the present one is that subjects were relatively unpracticed on the task in the former but well-practiced in the latter, and the main difference in results was that an rTMS effect was seen directly in the RT data of the former studies but not the latter. In terms of the present drift rate finding, these differences suggest that within the decision process, at least in a well-practiced state, the mechanism underlying the integration of congruent properties differs and can be more easily distinguished from that underlying the integration of incongruent properties. With practice, the effect of right parietal rTMS on automatic processing seen in the earlier studies may change such that it only occurs in the congruent state, while the decision process in the incongruent condition may rely more heavily on other brain regions, such as anterior cingulate cortex. A further study comparing the effects of rTMS on subjects that are task-naïve vs. practiced would be needed to verify this. Moreover, the diffusion model results and speculations based on them result need to be taken cautiously as the analysis was done post hoc and the study was not designed optimally to evaluate the drift. For example, it only included 48 trials per condition, while it has been demonstrated that about hundred trials are needed to estimate this parameter robustly [25]. In addition, other models might be tested: for example, the drift diffusion model used here fits performance data to a single processing stage, while a dual-stage model, involving an early evidence accumulation phase followed by a later decision phase might exhibit a more robust fit to the data than a single-stage model [26]. Future studies might also want to conduct a drift modeling approach on single trial fMRI to delineate the neurocircuitry associated with practice and congruency effects.

A primary methodological difference between the Sack et al. [5] and Cohen-Kadosh et al. [6] studies and the present one is that subjects were relatively unpracticed on the task in the former but well-practiced in the latter, and the main difference in results was that an rTMS effect was seen directly in the RT data of the former studies but not the latter. In terms of the present drift rate finding, these differences suggest that within the decision process, at least in a well-practiced state, the mechanism underlying the integration of congruent properties differs and can be more easily distinguished from that underlying the integration of incongruent properties. A further study comparing the effects of rTMS on subjects that are task-naïve vs. practiced would be needed to verify this. Success in such a study would result in a toolbox of manipulations involving states of practice, active and sham TMS applied at sites involved in task processing (e.g., left and right IPL, DLPFC), congruent and incongruent visual properties, and use of drift diffusion modeling. Use of this toolbox could provide the means for a more extensive investigation into the neural mechanisms of perceptual decision making.

## On replicating an rTMS effect observed in earlier studies

In this study, we tested whether fMRI-guided rTMS effects on numerical Stroop task found in Cohen Kadosh et al. [6] and Sack et al [5] could still be observed while using more advanced techniques that have emerged in the last decade. In general, we adhered to their reported

methods, and when possible, we substituted procedures that corresponded to more recent experimental standards (for example by employing electrical sham stimulation). We expected that the substitution for some of the older techniques used in the earlier papers would potentially improve the effect size they obtained. However, contrary to these expectations, we did not find better results, and instead failed to show any superiority of active rTMS overs sham stimulation. While it is clear that reproducing an experiment and its outcome in a different lab is never simple or even necessarily straight-forward, it is a useful exercise to examine the reasons that the replication of rTMS effect might not have occurred. Below we provide a list of the changes we made compared to the original studies and their possible consequences on the outcome of this study, as well as considerations for future rTMS studies using the numerical Stroop task to improve rigor and reproducibility. These are grouped into features involving the task itself, the study design, and rTMS delivery.

## Issues concerning handling of the Stroop task in experimental procedures

**Practice effect.** After a brief introductory period, the difference in RT between congruent and incongruent conditions is maintained in the numerical Stroop task, even after extensive training (see, for example, Fig 1), which makes the task a good candidate for use in a repeated measures design in rTMS experiments looking at stimulation effects across congruency conditions. However, first subjects must be introduced to and practice the task. In this study, we trained participants with one block of 48 task trials before the MRI (followed by performance of 4 blocks of the task while in the scanner), and then a full block of 72 practice trials in the second session, prior to rTMS, which led to a relatively stabilized reaction time for the experimental conditions. In contrast, Sack et al. [5] only familiarized participants with the task for a few trials until it was clear the participants understood the task instructions. This difference in practice may have contributed to the differences seen in the effects of TMS between the studies. It should be noted, however, that the fMRI-targeted condition in Sack et al. received more practice on the task (because of performing additional blocks of the task while in the scanner) than their other targeting methods. This discrepancy in the degree of practice on the task across targeting conditions in Sack et al. calls into question whether there is definitive evidence that individualized fMRI-guidance is the most effective way to target rTMS to modulate performance on the numerical Stroop. To make such a definitive statement, it would be important to equate the degree of practice on the task across the targeting conditions.

It may very well be that a brain in an unpracticed state, while it is still optimizing the processing needed to carry out the task, is more vulnerable to external stimulation than a brain in a practiced state, where the task processing is more stable and set and resilient to disruption. Further, our interpretation of the Sack et al. [5] and Cohen Kadosh et al. [6] rTMS effect on the numerical Stroop task performance was related to an addition-by-subtraction mechanism, where performance enhancement occurred because rTMS disrupted the interference of the irrelevant stimulus dimension (font size) in the decision process. However, our study using well-practiced subjects found that the application of rTMS did not lead to significant improvements in RT in the incongruent condition, suggesting that there may be a ceiling effect in terms of the potential benefits of rTMS to right IPS in enhancing RT in that condition of the Stroop task. Future studies examining the effect of rTMS on the numerical Stroop task in both practiced and unpracticed states may be warranted and could examine whether the locus of the effect shifts in practiced subjects to another location in the prefrontal-parietal executive network involved in the task.

Our findings, and their contrast with the previous studies using the numerical Stroop, also highlight the importance of considering individual performance levels, training protocols, and

potential ceiling effects when investigating the effects of TMS in cognitive tasks such as the numerical Stroop task. Further research is needed to explore the optimal conditions and target populations where rTMS interventions may have the greatest impact on task performance. Incorporating a staircase procedure in future studies may offer a viable solution to mitigate the confounding effects of training [27]. By dynamically adjusting the task difficulty based on individual performance, a staircase approach can help maintain a consistent level of challenge and control for potential learning effects. This can enhance the accuracy and reliability of assessing the impact of TMS on task performance and provide valuable insights into the underlying mechanisms of cognitive processes. Further research is needed to explore the feasibility and effectiveness of implementing a staircase procedure in this context.

**Use of performance feedback.** One specific manipulation we implemented in our task presentation was to provide the participants with feedback regarding their performance, which was not employed in the earlier studies. In our study, we decided to provide trial-by-trial feedback for the two practice blocks to facilitate learning, and feedback on average performance over a whole block of trials for each of the MRI and TMS blocks. Our decision was informed by Liu et al. [28] who used a Hebbian reweighting model to demonstrate that both types of feedback induce significant, equal learning.

**Accounting for the numerical distance between the digit stimuli.** Neither the present study nor Sack et al. [5] or Cohen Kadosh et al. [6] considered the interaction of the congruent/incongruent Stroop effect and the varying effect on performance caused by the "distance effect" in the contrast between the cardinality of the two stimulus digits presented in a given trial of the task, which potentially can add to variability in performance across individuals and work to obscure the sought after rTMS effect. Generally, the "distance effect" has been found to be an inverse relationship of RT and the cardinality difference between two numbers. For example, Moyer & Landauer (1967) [23]observed that participants were faster when the distance between numbers was greater (e.g., 2 and 7 versus 2 and 4), and that larger numbers (e.g., 8) were statistically more likely to be correct answers due to fewer single digits being greater than that number (e.g., 1-7 < 8<9). Methods to control for the distance effect include using number pairs with a set numerical ratio of 0.3 (e.g., 6, 2) [29], and presenting set pairs with differences of 1 (e.g., 3-4) and 5 (e.g., 4-9) to control the difference and number of times each digit was presented [30]. However, these controls limit the variability of trials and may lead to artificially low reaction times. While not accounted for in our study, the distance between the digit pairs used as stimuli should be balanced such that these effects are expected to be similar between active and sham conditions. Future studies using the numerical Stroop task should consider these limitations.

## Issues involving experimental design

**A possible cumulative rTMS effect.** The five subjects in the group using individualized fMRI targeting in Sack et al. [5] received sham and active rTMS conditions in the same session, and as a group showed a significant active-sham difference in numerical Stroop performance. In contrast, in the present study the participants who first received active rTMS followed by sham rTMS did not show the significant reaction time improvement in incongruent trials that was seen in the group that began with sham rTMS (in post hoc testing following a Stimulation x Congruency x Order interaction). While this could be an accident of sampling in groups with small sample size, it may reflect, instead, a potential carryover effect of rTMS. Online stimulation is expected to modulate neural activity in an acute fashion, providing insight into the timing of neural processing that underlies behavior, and there is an underlying assumption of independence between trials in the effect of rTMS. However, very few studies have tested

whether these short-term effects could accumulate over time. To our knowledge only two studies, including one from our group, have tested this assumption and concluded there is a lack of evidence for such cumulative rTMS effects [31, 32]. However, the stimulation parameters were different between each of these studies (10Hz during 3 seconds at 110% rMT over the DLPFC and superior parietal lobule; 5Hz during 4 seconds at 120% rMT over the DLPFC). These differences prevent direct comparison with each other or with this specific protocol. A potential way to assess this effect would be to interleave trials with and without stimulation. Overall, in terms of experimental design, we would recommend performing active and sham stimulation with sufficient time between the two conditions (e.g., perform them on different days) to mitigate any potential confound with accumulating effects.

**Electrical sham.** Sack et al. [5] used a placebo coil for sham stimulation, in which the sound of the TMS pulse and the pressure of the coil on the head are reproduced, but the sensation of the pulse is not. We used an electrical sham approach that uses electrodes applied to the participant's head underneath the TMS coil, that delivers a weak electrical current that mimics the somatosensory stimulation induced by active rTMS. We used a titration approach where we asked participants to report whether the sensation was tolerable. While this approach appears to be an improvement, it is also associated with disadvantages. For example, because of hair thickness on the parietal cortex, it was sometimes difficult to keeping sham electrodes firmly adhered to the scalp, resulting in sham TMS pulses not felt by some participants, even at the maximum stimulation intensity. It is also possible that the residual E-fields induced in the brain by sham electrodes lead to potential neuromodulatory effects on behavioral performance [19]. Further, to protect subject blinding, we followed the recommendations from Smith, & Peterchev (2018) and told the participants that two types of stimulation would be applied. However, since our study did not include any blinding questionnaire, we cannot conclude on whether participants felt any difference. To address these limitations, future studies could include post TMS questions on this topic. Since Sack et al. [5], did not use this sham methodology, potential differences in performance caused by the addition of electrical stimulation could be another reason contributing towards the failed replication.

## Issues involving TMS delivery

**Target selection.** In selecting a TMS target through the use an individual fMRI, the individual's peak activation in a contrast of interest within a circumscribed cortical region is chosen. But the actual size of that region is somewhat challenging to delimit *a priori*, as it is generally based on previous coordinates in a group analysis but should also consider spatial variability across individuals. We chose to follow the description given in Sack et al. [5]: the individual location within the right IPS exhibiting the strongest BOLD-signal contrast for the SCE. To select the TMS target, we used the IPS mask from the Neurosynth website (https://neurosynth.org/), within which we estimated the location the peak activation for each participant. However, the size of this mask was quite large and the choice of target within this mask suffers from a large variability across subjects in the location of the target selected, probably due to the lack of additional anatomical constraints in our mask (Fig 2B). We also constrained our choice to be the most active location situated on superficial cortex, to induce the strongest E-field and potentially lead to the strongest TMS efficacy. This clearly differed from Sack et al., who did not introduce such a constraint, and in fact chose targets that were deeper in cortex (but, it turned out, grouped closer together). While Sack et al. did not describe the coil orientation chosen for each target, in the present study the coil orientation was chosen such that the second phase of the induced E-field was perpendicular and directed into the nearest sulcal wall for each individual subject. This was done visually rather than through E-field modeling and

could have suffered some variability as a result. Overall, while there is general agreement in the field that, as concluded by Sack et al. [5], the use of individual fMRI activations for TMS targets is the best method for TMS target selection, given the low signal-to-noise levels in individual fMRI measures, there is still room for improvement.

A potential approach to overcome these limitations could be to rely on information from causally derived targets beyond the causal evidence provided by earlier TMS studies [5, 6]. For instance, studies have demonstrated that selecting targets from brain lesions or deep brain stimulation studies could potentially serve as effective TMS targets (see [33] for a review). While most causal targets have been derived from psychiatric symptoms such as depression or anxiety, examining behavioral deficits associated with focal brain injuries or deep brain stimulation can help selecting a TMS target for this specific executive function.

Regarding the group fMRI results, contrary to our expectations, while we were able to find significantly activated voxels for most of our participants (64%), our results at the group level do not indicate any significant activations in the right IPS in the Incongruent>Congruent contrast. Instead, we found some small clusters in the left IPS, and some larger clusters in subcortical structures such as the corpus callosum, and the cingulate cortex, the latter regions involved in conflict monitoring [34]. However, the lack of a group activation in the right IPS does not necessarily imply that it was not a good target. Indeed, it has been demonstrated that fMRI group analysis might not always detect the brain regions that are necessary at the individual level, and instead consider the individual topographic variability as noise [35] and that applying rTMS on subject-specific regions, even though variable, can lead to a significant rTMS effect while stimulating the group fMRI activation site does not [35].

A potential way to improve rTMS effects in this task might be to investigate the role of the left IPS during the Stroop task. To our knowledge, only two studies have tried this location and demonstrated either a stronger effect by applying TMS on the left IPS than the right IPS [36] or an effect only for the right hemisphere [6]. Another approach could be to conduct a connectivity analysis and to stimulate the superficial region showing the strongest connectivity with the corpus callosum. This has not been tested with this region and with this task, but this connectivity-based technique has previously demonstrated strong effects when rTMS was applied over the parietal cortex, to indirectly modulate the hippocampus during an episodic memory task [37] and it may be useful in the present context.

**Stimulation intensity.** We used the same dosing strategy of applying a fixed intensity across subjects that was used in the original Sack et al. study (60% maximum stimulator output). However, while the stimulators used in their study and the present one were the same models, the TMS coil we used (A/P-B65) was not the one used by Sack et al (B-70). The electric field generated by those two coils are slightly different [38], and that, combined with the more superficial location of our TMS targets compared to Sack et al., could also be a potential confound explaining the difference in outcomes. Further, individual differences in anatomy are something that should be taken into account when choosing intensity, given the correlation we found between the intensity expressed in relation to each individual's motor threshold and the size of the rTMS effect observed. The correlation suggests that using higher intensity might increase the rTMS effect.

**Number of pulses and stimulation timing.** Like Sack et al. [5], we delivered three pulses of 10Hz rTMS at 220, 320, and 420 ms post-stimulus onset. This timing was based on past event-related potentials (ERPs) studies that found ERP components correlating perceptual processing of congruity and numerical distance over the parietal electrodes between 300-480 ms [39]. Szucs and colleagues sought to determine whether the facilitation and interference effects of relevant/irrelevant information occurred during perceptual or response processing stages of the numerical Stroop task. To do so, they had participants conduct two sub-tasks,

either selecting the value greater in numeric value or physical size. Citing early work discussing the bidirectional congruency effects in the numerical Stroop task [40] and finding similar early ERP timing in both sub-tasks, they concluded that numerical and size dimensions of the task were processed in parallel regardless of relevance. With faster responses in the size task, they identified the facilitation and interference of the physical size on the numerical value determination. More recently, Huang et al. [41], investigated the interaction between task-relevant and irrelevant features on task performance, and found results contradicting Szucs & Soltez [38]. Huang et al. (2020) evaluated a temporally longer range of ERPs including the late positive complex (LPC). They found more positive responses in congruent than incongruent in early (320-520 ms) and late (520-720 ms) LPCs, with early also showing more positive responses in congruent vs neutral trials. Therefore, it is possible that the 220-420 ms was not the optimal timing for this task, and future studies might want to investigate different timing or add more pulses to cover a larger section of the neural processing.

*Summary of replication findings.* In the present study, we tried to reproduce the rTMS effect on numerical Stroop found in Sack et al. [5] and Cohen Kadosh et al. [6], with some newer methods that are now available in the field, such as robotic coil holder and electrical sham, hoping to find a similar or even stronger rTMS effect by combining those approaches with a larger sample size. In designing our experiment, we did not initially perform a power analysis on the optimal number of participants needed to observe significant difference between active and sham rTMS, relying on the findings of Cohen-Kadosh et al., and Sack et al., who each found a significant rTMS effect with 5 participants. While we tried to emulate their methods, the changes we did make in the experimental design may have potentially lowered the effect size, and we may have needed to employ a larger sample. There were some differences in the procedures used in the earlier and present studies. In terms of the numerical Stroop task, we practiced our subjects to a greater degree, additionally using feedback to accelerate their learning. While we succeeded in removing the confound of learning, we may have altered the neural processing involved, changing the sensitivity of right IPS to rTMS in the context of the Stroop task. Regarding experimental design, we differed in the addition of electrical stimulation to the sham condition, and we may also have found a cumulative effect of active/sham condition order that may have masked the performance effect we sought. Additionally, there were differences in how targeting choices using subject-specific fMRI data were handled that may have had profound effects on our results. We further identified several issues that were common to all the studies, involving stimulus choices, unbalanced distances between the digit stimuli presented, and choices of rTMS parameters such as intensity and stimulus timing, which could impact rTMS effects on the numerical Stroop task.

## Conclusions

In two earlier studies the use of rTMS to right IPS during performance of a numerical Stroop task resulted in strong RT changes reflecting a diminishment of the interference of irrelevant but automatically processed visual properties [5, 6]. While we did not directly replicate those findings, application of a drift diffusion model to performance data demonstrated a strong effect of rTMS on the drift rate variable in the decision process. This analysis revealed possible differences in how congruent and incongruent properties are integrated during perceptual decision making. To compare rTMS targeting methods during a cognitive task, we first must be able to validate and replicate the rTMS effect on the task. Our results show that replicating rTMS effects remains challenging and we highlight steps that can be taken to improve the rigor and reproducibility in rTMS studies of cognition. For example, the differences and issues in replicating rTMS studies that we discussed point to the need for more extensive

experimental planning and piloting, as well as more documentation of methodology (such as degree of practice on the task). Moreover, the sensitivity to changes of the effect of rTMS on the numerical Stroop task and its resistance to replication calls into question conclusions that can be drawn from it, and thus indirectly, conclusions on the best method for rTMS targeting. This calls for further work establishing best methods for TMS targeting. It also suggests more generally the need for further development of procedures to establish what constitutes a brain/behavior "TMS effect", both in terms of the modeling of performance data and on the cognitive processes involved, as well as engagement of brain networks supporting that performance. The application of more sophisticated behavioral performance analysis such as drift diffusion modeling may help parse out more subtle effects of rTMS on brain networks involved in the task.

## Supporting information

**S1 Table. Summary of stimulation target location and stimulation intensity for each participant.** Stimulation coordinates are in LPI coordinates system, t-scores at the stimulated target, stimulation intensity expressed as a percentage of the resting motor threshold, Euclidean distance to Sack et al. average target, and scalp-to-cortex distance for each participant. Highlighted in blue are the lowest values and in red the highest values. Participant is italics font was the outlier not included in the analyses.
(DOCX)

**S2 Table. Overlap between activated clusters in the Incongruent > Congruent contrast and their overlap with the Glasser atlas.**
(DOCX)

## Acknowledgments

We would like to thank Dr. Eudora Jones, Dr. Fahad Mukhtar, and Dr. William Regenold for providing medical coverage for each session; as well as Cristina Abboud, Megan Hynd, Elyssa Feuer, Paul Rohde, and Pei Robbins for their help with participants' screening.

This work utilized the computational resources of the NIH HPC Biowulf cluster (http://hpc.nih.gov).

## Author Contributions

**Conceptualization:** Lysianne Beynel, Zhi-De Deng, Bruce Luber, Sarah H. Lisanby.

**Data curation:** Lysianne Beynel, Hannah Gura, Zeynab Rezaee, Ekaete C. Ekpo.

**Formal analysis:** Lysianne Beynel, Hannah Gura, Zeynab Rezaee, Ekaete C. Ekpo, Zhi-De Deng.

**Funding acquisition:** Zhi-De Deng, Bruce Luber, Sarah H. Lisanby.

**Investigation:** Hannah Gura, Zeynab Rezaee, Ekaete C. Ekpo, Janet O. Joseph.

**Methodology:** Lysianne Beynel, Janet O. Joseph, Bruce Luber.

**Project administration:** Bruce Luber.

**Software:** Hannah Gura, Paul Taylor.

**Supervision:** Lysianne Beynel, Bruce Luber, Sarah H. Lisanby.

**Validation:** Bruce Luber.

**Visualization:** Zhi-De Deng, Paul Taylor.

**Writing – original draft:** Lysianne Beynel, Hannah Gura, Zeynab Rezaee, Ekaete C. Ekpo, Bruce Luber.

**Writing – review & editing:** Zhi-De Deng, Janet O. Joseph, Paul Taylor, Sarah H. Lisanby.

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
