## [Decision Letter · Decision Letter 0]

1 Feb 2024

PONE-D-23-36866What is the best way to target rTMS?  Lessons learned from a replication attempt on fMRI-guided rTMS modulation of the numerical Stroop task.PLOS ONE

Dear Dr. Beynel,

Thank you for submitting your manuscript to PLOS ONE. After careful consideration, we feel that it has merit but does not fully meet PLOS ONE’s publication criteria as it currently stands. Therefore, we invite you to submit a revised version of the manuscript that addresses the points raised during the review process.

The reviewers clearly see value in the work but also there is some consistency that the title is misleading given no head-to-head comparison of targeting approaches. The reviewers make a number of other excellent points, requests for additional analyses, and suggestions for framing that will further improve the manuscript.

We look forward to receiving your revised manuscript.

Kind regards,

Desmond J. Oathes

Academic Editor

PLOS ONE

 [Beynel L., Gura, H., Rezaee, Z., Ekpo, E., Joseph, J., Deng, Z-D., Luber, B., and Lisanby, S.H. are supported by the NIMH Intramural Research Program (ZIAMH002955).

Taylor, P. was supported by the NIMH Intramural Research Program (ZICMH002888) of the NIH/HHS, USA.].  

3. PLOS requires an ORCID iD for the corresponding author in Editorial Manager on papers submitted after December 6th, 2016. Please ensure that you have an ORCID iD and that it is validated in Editorial Manager. To do this, go to ‘Update my Information’ (in the upper left-hand corner of the main menu), and click on the Fetch/Validate link next to the ORCID field. This will take you to the ORCID site and allow you to create a new iD or authenticate a pre-existing iD in Editorial Manager. Please see the following video for instructions on linking an ORCID iD to your Editorial Manager account: " ext-link-type="uri" xlink:type="simple">https://www.youtube.com/watch?v=_xcclfuvtxQ".

[The opinions expressed in this article are the authors' and do not reflect the views of the National Institutes of Health, the Department of Health and Human Services, or the United States government.  Dr. Deng is inventor on patents and patent applications on electrical and magnetic brain stimulation therapy systems held by the National Institutes of Health (NIH), Columbia University, and University of New Mexico. Dr. Lisanby is inventor on patents and patent applications on electrical and magnetic brain stimulation therapy systems held by the NIH and Columbia University, with no remuneration.]. 

5. Thank you for uploading your study's underlying data set. Unfortunately, the repository you have noted in your Data Availability statement does not qualify as an acceptable data repository according to PLOS's standards.

6. Please include the reference section of your manuscript.

7. We note that Figure(s) 4A and 4B in your submission contain copyrighted images. All PLOS content is published under the Creative Commons Attribution License (CC BY 4.0), which means that the manuscript, images, and Supporting Information files will be freely available online, and any third party is permitted to access, download, copy, distribute, and use these materials in any way, even commercially, with proper attribution. For more information, see our copyright guidelines: http://journals.plos.org/plosone/s/licenses-and-copyright.

a. You may seek permission from the original copyright holder of Figure(s) 4A and 4B to publish the content specifically under the CC BY 4.0 license. 

Reviewers' comments:

Reviewer's Responses to Questions

**Comments to the Author**

1. Is the manuscript technically sound, and do the data support the conclusions?

Reviewer #1: Yes

Reviewer #2: Yes

Reviewer #3: Yes

Reviewer #4: Yes

Reviewer #5: Yes

2. Has the statistical analysis been performed appropriately and rigorously? 

Reviewer #1: Yes

Reviewer #2: Yes

Reviewer #3: Yes

Reviewer #4: Yes

Reviewer #5: Yes

3. Have the authors made all data underlying the findings in their manuscript fully available?

Reviewer #1: Yes

Reviewer #2: Yes

Reviewer #3: Yes

Reviewer #4: Yes

Reviewer #5: Yes

4. Is the manuscript presented in an intelligible fashion and written in standard English?

Reviewer #1: Yes

Reviewer #2: Yes

Reviewer #3: Yes

Reviewer #4: Yes

Reviewer #5: Yes

5. Review Comments to the Author

Reviewer #1: I really enjoyed reading this paper. The authors did a fantastic and rally went the extra mile in terms of methodology and data analyses. Chapeau! I also love that they were inspired by our original article from 2009, and aimed at replicating our comparison of different TMS coil positioning approaches.

The discussion is comprehensive and addresses many factors that may have contributed to the result(s) and may explain differences between their and our own findings from 2009.

There is so much food for thought in there that I absolutely applaud such as the question of replicability and robustness of TMS-induced behavioral performance effects and their dependence on, e.g., degree of practice on the task at hand, or experimental design choices such as between versus within-subject comparisons etc. This is all wonderful.

My only and main concern here is that the title, abstract and introduction, in fact the overall framing of this paper, is misleading and unjustified. The title literally promises to address the question “What is the best way to target rTMS?”

And then the abstract and introduction frames this as a replication of our comparison of different targeting approaches from 2009. But, in sharp contrast to this framing , in the current study no comparison of targeting approaches is presented. In fact, only fMRI-guided TMS neuronavigation is used to test for effects on performances in the numerical stroop task. The current study only compares active and sham stimulation and fails to replicate the behavioral RT effects reported in earlier TMS studies using the same task. There is no comparison of TMS coil positioning approaches and the question in the title “What is the best way to target rTMS?” is not addressed by the study. I understand that replicating the behavioral effect is a step towards a replication study of the localization differences reported by us, but now readers will get the impression that they already did that and failed.

The authors seem to suggest that the failed replication of the behavioral effect using fMRI-guided TMS casts doubt on the localization / TMS targetting effect. I think this conclusion is very questionable.

The main gist and focus of our study in 2009 were to compare the relative differences between different TMS targeting approaches. It was not meant to claim that you need only 5 subjects or 15 subjects or whatever number of subjects to find a significant behavioral finding in the numerical stroop test when using this or that targeting approach. The design and overall extremely low sample size was completely unsuited for that (5 subjects, between subject design). We were also not interested in the numerical stroop task per se. This task just served as a tool, a means to compare the performance differences induced by the various TMS targeting approaches we compared with each other, showing relative differences. The numerical stroop task we used was brought to us by our visiting researcher, Dr. Roi Cohen Kadosh, who was PI of a research line using this task in the fMRI.

In related publications, e.g. Cohen Kadosh 2007, this team reported right parietal activity changed during the execution of this numerical stroop task and behavioral effects when stimulating those individual right parietal activity clusters using TMS.

I understand that the authors originally wanted to replicate our targeting approach study but got stopped at step 1, when hoping to first find the expected behavioral changes when using the supposedly best targeting approach: individual fMRI-guided TMS. Since they could not find any behavioral effects here, they now publish this still in the framing of an intended and failed replication of our TMS targeting study. But in fact, they failed to find the behavioral effects using one targeting approach. This is something completely different.

In fact, when looking at their results in more detail, the issue of replicating previous work already starts with the fMRI data itself. Unlike previous work by Cohen Kadosh and others, the fMRI data of the current does not demonstrate clear individual right parietal activity clusters to underlie the execution of this task. In fact, in many cases, their fMRI data, if really used to localize the individual hotspot of this task for TMS targeting, would have suggested areas in the left parietal cortex.

So, ironically, because they aimed to replicate our study of 2009, they wanted to stick to the right IPS (because this is where we fund our fMRI hotspot in each participant for this task), but by doing so, they actually violated the entire principle of individual fMRI-guided TMS targeting which bases the TMS positioning on the individual fMRI hotspot, data-driven, not hypothesis- or replication-driven.

The should have listed t the individual fMRI findings and stimulate at the fMRI0based hotspot, and maybe, if they then would have found a significant behavioral effect, the could have compared this to other targeting approaches. This would have been closer to replicating our study than sticking to a right IPS cluster because of an a priori principle. Redo the analyses, without masks or restrictions and do the TMS based on the individual hotspot, no matter whether right or left IPS. Otherwise, you are not capitalizing on the power of individual fMRI-guided TMS.

In sum, great study, many lessons learned, great discussion, but title, and framing as a replication of our study from 2009 is misleading, and the study is really not a replication of our work as it differs in too many methodological aspects and doesn’t even include any comparison of different TMS targeting approaches. This can easily be fixed by changing title and reframing abstract and introduction.

But I want to again congratulate the authors for conducting this study, I want to encourage them to continue on this path, and express my admiration for their high scientific standards and methodological rigor.

Best wishes,

Alexander Sack

Reviewer #2: The present study attempted to replicate prior findings showing that short bursts of high frequency TMS delivered to the right intraparietal sulcus disrupt the Stroop effect (i.e., Cohen Kadosh et al., 2007; Sack et al., 2009). Well conducted replication efforts like this one are rare, at least as can be determined by the published literature. However, I have several questions, comments, and suggested revisions that I think could strengthen the paper.

(1) Was a power analysis performed? If not, the authors should justify their sample size and acknowledge the lack of a power analysis as a limitation. I am skeptical of the authors’ claim that “rTMS effects on task performance are typically seen using sample sizes of thirteen and up”. This claim was made on the basis of their review of a heterogeneous behavioral TMS literature using a variety of TMS protocols and cognitive tasks. Surely, protocol and task differences impact power considerations.

(2) Was the decision to perform drift diffusion models made post-hoc? If so, consistent with the spirit of reproducibility that motivated the present study, it should probably be described that way. I was able to find general study details on clinical trials.gov but did not see a description of planned analyses.

(3) The Discussion includes material that should probably be outlined in the Introduction and better detailed in Methods. For example, drift diffusion models are mentioned only briefly in the Introduction. It would be useful to briefly outline why this approach was used versus others. More importantly, as described in detail in the Discussion section, there are several ways in which the present study differs from the studies that it sought to replicate. It would be useful to preview what motivated these differences in the Introduction and cover them in more detail in Methods.

(4) I think it would be worth testing to see if the ANOVA homoscedasticity assumption is violated, as the congruent/incongruent Stroop conditions may have different variances. If this assumption is not met, mixed models would be a more appropriate way to analyze these data.

(5) It may be worth mentioning (although I think this should be up to the authors) that causally derived targets may be better candidates than fMRI correlates, as these correlates may be spurious or reflect compensatory processes. TMS accessible cortical areas that are strongly connected to TMS/DBS sites or lesions that alter Stroop performance or Stroop relevant processes may be particularly promising.

(6) Could the authors please include a Methods figure detailing study procedures by visit?

(7) I am curious about the authors’ speculations on their drift rate finding that TMS affected congruent versus incongruent trials. This seems worth briefly commenting on in the Discussion section. However, I leave this up to the authors if they feel the speculations are too speculative.

Reviewer #3: This paper conducts a replication study of the facilitatory effects of rTMS on numerical Stroop task performance. They employed personalized targeting of the right intraparietal sulcus, as determined by peak activation. Although they were not able to replicate the reaction time results, they did find an increase in the drift rate in the congruent trials. The failure to replicate was attributed to various methodological differences, including task and stimulation administration.

Overall, the paper is clearly and concisely written and the included figures illustrate the results in a simple and elegant way. I also applaud the authors for a replication attempt due to the profound lack of these studies. My main point of criticism is the employed statistical methods. The use of more robust statistical tools could give more confidence in the results by appropriately modeling the experimental structure. I am aware that this suggestion brings the scope of this study slightly past replication, and note that it already involves an innovative use of DDM. I believe the authors can enhance their already meaningful contribution to the literature.

Abstract:

- No comments

Introduction:

- #112: Please further motivate the decision to include the drift diffusion model (DDM) in the analyses. While I am aware that these models are highly influential and agree that they are interesting in this context, the theoretical reason to use it was not clear. I believe there are compelling reasons to do so, but they should be stated even if the authors did not have any clear or directional hypotheses.

- I also think that while the Stroop task is quite old and famous, the deck is stacked against some studies due to the relatively small and variable behavioral effect. The authors do discuss these issues, but I do think a bit more could be stated about using traditional TMS sequences to find small and noisy effects in general. For instance, is there a confidence interval in which we would expect the current and previous study’s results to fall?

Methods:

- Was motion confound regression applied to the BOLD time series? If so, please indicate which motion parameters were used.

- Did the authors consider that the observations are not independent, as many samples are derived from the same participant? The use of a linear model, such as an ANOVA may not be appropriate in this case and may warrant the use of more advanced statistical methods such as a linear mixed effects model.

- For the DDM, although 5% of the trials were modelled with a different distribution, were outlier reaction times ( 200ms) removed from the analyses? It is likely that these trials reflect a different underlying cognitive process than what is trying to be measured.

- There are concerns that the number of trials used for each of the DDMs may have been too low, which may have affected the precision of parameter estimations. Please refer to Lerche et al. 2017 and comment with regards to the current study.

Results:

- As the lack of a size congruency effect was one of the main negative findings in the replication, please include a figure for the demonstrating the lack of a size congruency effect.

Discussion:

- There have been multiple studies involving DDM in the context of attentional and cognitive control that have required modifications to the DDM to account for these types of data (Hübner et al., 2010). Please refer to these works in the limitations section.

- Although the authors use the DDM results to discount the role of the right IPL in automatic spatial magnitude processing, do they have any cognitive or neurobiological explanation for why the drift rate may have increased in only the congruent condition?

- The choice of fMRI preprocessing parameters can change the findings of otherwise identical studies. Please comment on the differences in preprocessing schemes and how they may have affected your results.

References:

Lerche, V., Voss, A., Nagler, M. (2017). How many trials are required for parameter estimation in diffusion modeling? A comparison of different optimization criteria. Behavior Research Methods, 49(2), 513–537. https://doi.org/10.3758/s13428-016-0740-2

Hübner, R., Steinhauser, M., Lehle, C. (2010). A dual-stage two-phase model of selective attention. Psychological Review, 117(3), 759–784. https://doi.org/10.1037/a0019471

Minor:

- #213: Was the intention “highest tolerable intensity”?

Reviewer #4: General comments:

Overall this is a valuable study for the cognitive TMS field. It highlights the many factors that influence whether a TMS effect on behaviour will be observed. The discussion is particularly strong in this regard.

The study is primarily posed, however, as a replication of Sack et al. (2008), looking at the efficacy of individualised fMRI targeting to modulate numerical Stroop task performance. The aim of the Sack study was to compare different targeting approaches, whereas here the aim was to replicate one of the results from Sack et al. My first suggestion is therefore to change the title of the paper, as the current study is not addressing the targeting issue directly. (If replication of the targeting approach was the main aim of the present study, then this would have been better achieved by comparing against other targeting approaches, as per the original study.) While the finding of targeting approach (individual v group fMRI clusters for example) might not have been directly replicated, this issue has come up in other publication (e.g., Feredoes et al., 2007, J Neurosci), and many have used an individualised fMRI approach and shown individual variability around a specific target. Alongside findings from e-field modelling, it’s not clear this is such an issue these days.

While replications, especially in the cognitive TMS field, are always extremely valuable, the ‘message’ of this study is more related to the number of considerations that a researcher must face when designing and conducting study on behaviour. I would therefore suggest re-framing the paper along these lines, through an attempt to replicate what could be regarded as a robust finding, given the reliability of the Stroop effect and individualised targeting. I feel that this would be of interest to the field, given that the targeting of cognitive processes with TMS, while conceptually simple, remains challenging, and the targeting approach is only one consideration when it comes to maximising TMS efficacy.

Additional comments:

- How were the contrast maps thresholded? The t-values of the stimulated targets in Table 1 seem on the low side?

- Could some more detail be provided on how the IPS target was selected? The methods states that the ‘statistical peak’ within the right IPS was chosen (line 191), but what if there was more than one peak? Was it the cluster with the highest t-value? Were there any further anatomical constraints beyond being within the Neurosynth mask? From Figure 4 it does look like the individual targets are anatomically variable, which is not an issue by itself, but some of the targets look like they are in the precuneus, and several others are in the inferior parietal cortex.

- Given the multiple parameters that comprise a TMS study, would it be worthwhile trying to combine as many of these as possible (since they were recorded) to see if they can predict an individual’s results? Currently the values from Table 1 were assessed as single correlations with behaviour. Could they instead be predictors in a regression model? (The distance to the Sack group coordinate might not be useful to include, however, as they did not demonstrate a significant effect of TMS when this was targeted.) Such an analysis would also fit with the aim of the paper, which is to highlight the need for more sophisticated analyses, but also it would fit with our understanding of individual differences in response to TMS.

- For the group-level fMRI analysis, the lack of an effect in right IPS is not necessarily an issue, and would speak to the anatomical variability of this region. As mentioned above, this issue has been covered by others such as Feredoes et al. It may be worth highlighting this particular point in the discussion when mentioning the potential of left IPS as a better target.

Reviewer #5: I am reviewing “What is the best way to target TMS? Lessons learned from a replication attempt on fMRI-guided rTMS modulation of the numerical Stroop task.” by Beynel et al. In this study, the authors sought to replicate results from the influential Sack et al. 2009 paper, which showed the relative advantage and increase in statistical power obtained when using individualized vs group-MRI based (or scalp based) TMS targeting for improving performance in a numeric Stroop task. The current study largely failed to replicate Sack et al., and thoughtfully acknowledged a few reasons why that might be, while making recommendations for future studies. I appreciated the overall effort and study, and thought the manuscript was thoughtful and well written. However, some of the results, design and analysis choices, raised questions and a few concerns for me, which are delineated below:

Major

1. Most strikingly, from the way the intro was framed, I was under the impression that the authors were going to directly compared contrast different TMS targeting strategies (e.g. anatomical vs. functional based) within subjects (in contrast with the between-subjects approach used by Sack et al.) However, the study design only included the individualized TMS targeting condition, as opposed to the critical comparison of e.g. MRI/group- based coordinates vs. individualized, functional-based coordinates. While this point might appear moot in the context of the current (null) results, I still think this point (i.e., the primary study aim, which is not exactly a replication of Sack et al., which would have required the aforementioned explicit targeting strategy comparison) should be recontextualized and clarified in the Intro (and title).

2. A critical aspect of the results that should be taken into account explicitly across all analysis (and it currently isn’t) is the 3-way interaction with TMS order (sham first vs control first). If I followed the results correctly, the result obtained with the group who had sham first is precisely the predicted result. Given that TMS (sham vs. active) order clearly matters here, all subsequent analysis should include order in the model as either a covariate or perhaps more ideally as an interacting factor (including the SCE analysis!).

3. Relatedly, how does overall Stroop performance compare between Sack et al. the current study? I am finding that RT differences for Congruent vs. Incongruent were slightly larger in the Sack study (104ms, potentially providing a better range to modulate those effects) vs the current study (71ms?), but I am not finding the average accuracy data in the Sack study... That would be important to know and compare across the samples.

4. And how does the age of the participants compare? (I was struck by the relatively older age of the current sample, 39 y old). I am revisiting the Sack paper, and seeing that the mean age was 24 y old in that study… This point relates back to the performance point above, particularly if performance differs across the 2 samples…

- Does age correlate with Stroop performance and/or with the magnitude of TMS effects found here?

5. How far apart were the MRI and TMS sessions….? Does the interval matter in terms of the magnitude of the effects?

6. Accuracy seemed quite a bit lower for the TMS day (88%) compared to the fMRI day (98%). Was that a significant difference between those days? If so, is it possible that TMS is impacting accuracy (and/or differing by TMS site)?

7. I was wondering about whether there is a chance that the ‘sham’ condition in this study was less of a sham than what was expected…. which is a point the authors themselves raised in the Discussion. Is there a way to more directly get at this question, e.g., by comparing stroop performance (accuracy or RT) in the sham condition vs. the baseline fMRI day?

8. I was not entirely clear on the interpretation of Figure 3, particularly regarding the specificity to the Congruent condition. Can that interpretation be more clearly spelled out, both as a short summary in the Results section, as well as in the Discussion section?

Minor

1. I found Figures 2 3 hard to parse as the Stroop effect is most easily visualized by comparing Congruent vs. Incongruent conditions directly, which are currently plotted far away from one another; wouldn’t plotting Congruent vs. Incongruent side-by-side, and Sham vs. TMS (etc) on the X axis, make those plots easier to read?

6. PLOS authors have the option to publish the peer review history of their article (what does this mean?). If published, this will include your full peer review and any attached files.

Reviewer #1: **Yes: **Alexander Sack

Reviewer #2: No

Reviewer #3: **Yes: **John D. Medaglia

Reviewer #4: No

Reviewer #5: No

---

## [Author Response · Author response to Decision Letter 0]

28 Feb 2024

¬¬Journal requirements:

Response: The manuscript has been edited to follow Plos one style templates.

 [Beynel L., Gura, H., Rezaee, Z., Ekpo, E., Joseph, J., Deng, Z-D., Luber, B., and Lisanby, S.H. are supported by the NIMH Intramural Research Program (ZIAMH002955).

Taylor, P. was supported by the NIMH Intramural Research Program (ZICMH002888) of the NIH/HHS, USA.]. 

Response: This information has been updated in our cover letter: “The funders had no role in study design, data collection and analysis, decision to publish, or preparation of the manuscript

3. PLOS requires an ORCID iD for the corresponding author in Editorial Manager on papers submitted after December 6th, 2016. Please ensure that you have an ORCID iD and that it is validated in Editorial Manager. To do this, go to ‘Update my Information’ (in the upper left-hand corner of the main menu), and click on the Fetch/Validate link next to the ORCID field. This will take you to the ORCID site and allow you to create a new iD or authenticate a pre-existing iD in Editorial Manager. Please see the following video for instructions on linking an ORCID iD to your Editorial Manager account: https://www.youtube.com/watch?v=_xcclfuvtxQ".

Response: The corresponding author ORCID ID has been added to the Editorial Manager

[The opinions expressed in this article are the authors' and do not reflect the views of the National Institutes of Health, the Department of Health and Human Services, or the United States government. Dr. Deng is inventor on patents and patent applications on electrical and magnetic brain stimulation therapy systems held by the National Institutes of Health (NIH), Columbia University, and University of New Mexico. Dr. Lisanby is inventor on patents and patent applications on electrical and magnetic brain stimulation therapy systems held by the NIH and Columbia University, with no remuneration.]. Please confirm that this does not alter your adherence to all PLOS ONE policies on sharing data and materials, by including the following statement: ""This does not alter our adherence to PLOS ONE policies on sharing data and materials.” (as detailed online in our guide for authors http://journals.plos.org/plosone/s/competing-interests). If there are restrictions on sharing of data and/or materials, please state these. Please note that we cannot proceed with consideration of your article until this information has been declared. Please include your updated Competing Interests statement in your cover letter; we will change the online submission form on your behalf.

Response: We have updated the competing interest’s statement in our cover letter. 

5. Thank you for uploading your study's underlying data set. Unfortunately, the repository you have noted in your Data Availability statement does not qualify as an acceptable data repository according to PLOS's standards. At this time, please upload the minimal data set necessary to replicate your study's findings to a stable, public repository (such as figshare or Dryad) and provide us with the relevant URLs, DOIs, or accession numbers that may be used to access these data. For a list of recommended repositories and additional information on PLOS standards for data deposition, please see https://journals.plos.org/plosone/s/recommended-repositories.

Response: Data have been added to “figshare” repository with this URL: https://figshare.com/account/home#/projects/194468

6. Please include the reference section of your manuscript. 

Response: References were at the end of the manuscript. We will add it separately in the document upload.

7. We note that Figure(s) 4A and 4B in your submission contain copyrighted images. All PLOS content is published under the Creative Commons Attribution License (CC BY 4.0), which means that the manuscript, images, and Supporting Information files will be freely available online, and any third party is permitted to access, download, copy, distribute, and use these materials in any way, even commercially, with proper attribution. For more information, see our copyright guidelines: http://journals.plos.org/plosone/s/licenses-and-copyright. We require you to either (1) present written permission from the copyright holder to publish these figures specifically under the CC BY 4.0 license, or (2) remove the figures from your submission:

a. You may seek permission from the original copyright holder of Figure(s) 4A and 4B to publish the content specifically under the CC BY 4.0 license. We recommend that you contact the original copyright holder with the Content Permission Form (http://journals.plos.org/plosone/s/file?id=7c09/content-permission-form.pdf) and the following text:“I request permission for the open-access journal PLOS ONE to publish XXX under the Creative Commons Attribution License (CCAL) CC BY 4.0 (http://creativecommons.org/licenses/by/4.0/). Please be aware that this license allows unrestricted use and distribution, even commercially, by third parties. Please reply and provide explicit written permission to publish XXX under a CC BY license and complete the attached form.” Please upload the completed Content Permission Form or other proof of granted permissions as an ""Other"" file with your submission. In the figure caption of the copyrighted figure, please include the following text: “Reprinted from [ref] under a CC BY license, with permission from [name of publisher], original copyright [original copyright year].”

Response: Figure 4 is our own figure, no copyright needed.

Reviewer #1: 

I really enjoyed reading this paper. The authors did a fantastic and rally went the extra mile in terms of methodology and data analyses. Chapeau! I also love that they were inspired by our original article from 2009 and aimed at replicating our comparison of different TMS coil positioning approaches. The discussion is comprehensive and addresses many factors that may have contributed to the result(s) and may explain differences between their and our own findings from 2009. There is so much food for thought in there that I absolutely applaud such as the question of replicability and robustness of TMS-induced behavioral performance effects and their dependence on, e.g., degree of practice on the task at hand, or experimental design choices such as between versus within-subject comparisons etc. This is all wonderful.

Response: Thank you very much (Merci!) Dr. Sack for those good words on our manuscript, we are glad that you found it interesting and, in the section below we provide a point-by-point response to your concerns.

1. My only and main concern here is that the title, abstract and introduction, in fact the overall framing of this paper, is misleading and unjustified. The title literally promises to address the question “What is the best way to target rTMS?” And then the abstract and introduction frames this as a replication of our comparison of different targeting approaches from 2009. But, in sharp contrast to this framing, in the current study no comparison of targeting approaches is presented. In fact, only fMRI-guided TMS neuronavigation is used to test for effects on performances in the numerical stroop task. The current study only compares active and sham stimulation and fails to replicate the behavioral RT effects reported in earlier TMS studies using the same task. There is no comparison of TMS coil positioning approaches and the question in the title “What is the best way to target rTMS?” is not addressed by the study. I understand that replicating the behavioral effect is a step towards a replication study of the localization differences reported by us, but now readers will get the impression that they already did that and failed.

Response: Thank you for raising this important concern. We acknowledge that our initial framing may have created a misleading impression regarding the scope of our study. While our ultimate goal with this research is to compare the efficacy of new targeting approaches, as you say we first wanted to ensure we could produce a reliable behavioral effect on the numerical Stroop task using TMS. Consequently, we decided to rewrite our manuscript so that the framing reflects the scope and methodology of our research. 

The new title is: “Lessons learned from an fMRI-guided rTMS study on performance in a numerical Stroop task.”

The abstract now reads: “The Stroop task is a well-established tool to investigate the influence of competing visual categories on decision making. Neuroimaging as well as rTMS studies have demonstrated the involvement of parietal structures, particularly the intraparietal sulcus (IPS), in this task. Given its reliability, the numerical Stroop task was used to compare the effects of different TMS targeting approaches by Sack and colleagues, who elegantly demonstrated the superiority of individualized fMRI targeting. We performed the present study to test whether fMRI-guided rTMS effects on numerical Stroop task performance could still be observed while using more advanced techniques that have emerged in the last decade (e.g., electrical sham, robotic coil holder system, etc). To do so we used a traditional reaction time analysis and we performed, post-hoc, a more advanced comprehensive drift diffusion modeling approach. Fifteen participants performed the numerical Stroop task while active or sham 10 Hz rTMS was applied over the region of the right intraparietal sulcus (IPS) showing the strongest functional activation in the Incongruent Congruent contrast. This target was determined based on individualized fMRI data collected during a separate session. Contrary to our assumption, the classical reaction time analysis did not show any superiority of active rTMS over sham, probably due to confounds such as potential cumulative rTMS effects, and the effect of practice. However, the modeling approach revealed a robust effect of rTMS on the drift rate variable, suggesting differential processing of congruent and incongruent properties in perceptual decision-making, and, more generally, illustrating that more advanced computational analysis of performance data can elucidate the effects of rTMS on the brain where simpler methods may not.” (PAGE 2)

We modified the introduction as follows: “Sack et al. (2009) concluded that the most effective approach was individualized targeting based on the task-related fMRI. This was an important result for the brain stimulation field in general, as one of the main moderators of rTMS efficacy is the targeting approach (Beynel et al., 2019). More than a decade after this elegant demonstration of the importance of individualized fMRI targeting, experimental TMS methodologies have advanced, with for example the development of robotic coil holders that allow for more precise and stable positioning of the coil relative to the stimulation target, and an electrical sham technique that mimics the TMS-induced sensations, and thus potentially better blinds subjects to experimental conditions. In the present study, we tested whether the addition of these techniques combined with an individualized fMRI-based targeting approach endorsed by Sack and colleagues could improve the rTMS effect on numerical Stroop performance, with the expectation based on Cohen Kadosh et al. (2007) and Sack et al. (2009) that rTMS applied over the IPS would decrease the interference of incongruent stimuli and thus lessen of the reaction time slowing in that condition. Additionally, contrary to the prior study from Sack et al., participants completed the task multiple times preceding rTMS application (one block of trials before the MRI session, 4 blocks during MRI, and one block in the TMS session prior to TMS), to stabilize performance and minimize variability due to practice effects. This approach enabled us to probe the influence of task practice on behavioral performance and fMRI activations. (PAGE: 5-6)

2. The authors seem to suggest that the failed replication of the behavioral effect using fMRI-guided TMS casts doubt on the localization / TMS targeting effect. I think this conclusion is very questionable. The main gist and focus of our study in 2009 were to compare the relative differences between different TMS targeting approaches. It was not meant to claim that you need only 5 subjects or 15 subjects or whatever number of subjects to find a significant behavioral finding in the numerical stroop test when using this or that targeting approach. The design and overall extremely low sample size was completely unsuited for that (5 subjects, between subject design). We were also not interested in the numerical stroop task per se. This task just served as a tool, a means to compare the performance differences induced by the various TMS targeting approaches we compared with each other, showing relative differences. The numerical stroop task we used was brought to us by our visiting researcher, Dr. Roi Cohen Kadosh, who was PI of a research line using this task in the fMRI. In related publications, e.g. Cohen Kadosh 2007, this team reported right parietal activity changed during the execution of this numerical stroop task and behavioral effects when stimulating those individual right parietal activity clusters using TMS. I understand that the authors originally wanted to replicate our targeting approach study but got stopped at step 1, when hoping to first find the expected behavioral changes when using the supposedly best targeting approach: individual fMRI-guided TMS. Since they could not find any behavioral effects here, they now publish this still in the framing of an intended and failed replication of our TMS targeting study. But in fact, they failed to find the behavioral effects using one targeting approach. This is something completely different. In fact, when looking at their results in more detail, the issue of replicating previous work already starts with the fMRI data itself. Unlike previous work by Cohen Kadosh and others, the fMRI data of the current does not demonstrate clear individual right parietal activity clusters to underlie the execution of this task. In fact, in many cases, their fMRI data, if really used to localize the individual hotspot of this task for TMS targeting, would have suggested areas in the left parietal cortex. So, ironically, because they aimed to replicate our study of 2009, they wanted to stick to the right IPS (because this is where we fund our fMRI hotspot in each participant for this task), but by doing so, they actually violated the entire principle of individual fMRI-guided TMS targeting which bases the TMS positioning on the individual fMRI hot

---

## [Decision Letter · Decision Letter 1]

10 Apr 2024

Lessons learned from an fMRI-guided rTMS study on performance in a numerical Stroop task

PONE-D-23-36866R1

Dear Dr. Beynel,

Congratulations, guys!

We’re pleased to inform you that your manuscript has been judged scientifically suitable for publication and will be formally accepted for publication once it meets all outstanding technical requirements.

Kind regards,

Desmond J. Oathes

Academic Editor

PLOS ONE

Additional Editor Comments (optional):

Reviewers' comments:

Reviewer's Responses to Questions

**Comments to the Author**

1. If the authors have adequately addressed your comments raised in a previous round of review and you feel that this manuscript is now acceptable for publication, you may indicate that here to bypass the “Comments to the Author” section, enter your conflict of interest statement in the “Confidential to Editor” section, and submit your "Accept" recommendation.

Reviewer #1: All comments have been addressed

Reviewer #2: All comments have been addressed

Reviewer #4: All comments have been addressed

2. Is the manuscript technically sound, and do the data support the conclusions?

Reviewer #1: Yes

Reviewer #2: Yes

Reviewer #4: Yes

3. Has the statistical analysis been performed appropriately and rigorously? 

Reviewer #1: Yes

Reviewer #2: Yes

Reviewer #4: Yes

4. Have the authors made all data underlying the findings in their manuscript fully available?

Reviewer #1: Yes

Reviewer #2: Yes

Reviewer #4: Yes

5. Is the manuscript presented in an intelligible fashion and written in standard English?

Reviewer #1: Yes

Reviewer #2: Yes

Reviewer #4: Yes

6. Review Comments to the Author

Reviewer #1: The authors did a very good job in addressing my previous comments.

I just want to share some additional thoughts I have regarding their assumption that selecting the right IPS as an individual target for fMRI-guided TMS is the only logical choice here based on previous work (Cohen kadosh and sack et al) , and thus independent of the actual individual fMRI data collected in the present study ( apriori focus on right IPS).

The failed replication of this right IPS dominance (in fMRI already is importnat to acknowledge. But the manuscript still focusses very much on the replication attempt of the behavioural effects when stimulating right IPS during the numerical stroop task. it assumes that everything is according to plan up until the lack of finding / replicating this behavioural effects when following the procedure as described in Sack et al (1).

But the actual difference (or problem) already starts with the fMRI data and thus before the actual TMS study. Unlike Cohen Kadosh, the current fMRI data reveals something different, namely an initial bilateral, right dominant IPS activation during the task, followed by a practise-dependent shift / reduction of the right IPS activation. This means that in the actual TMS study, many of the tested subjects showed no right IPS activation during task execution (maybe because of practise , maybe because the right IPS functional dominance is in itself not a reliable and robust finding).

Before testing the reliability and replicability of TMS target approaches (ultimate goal of this study), and even before testing the replicability of behavioural effects of right IPS TMS in numerical scoop task (as framed now), one may just as well question and test the replicability of the assumed right parietal functional activation during task execution.

I am no expert in this field of research but if Cohen-Kadosh found bilateral parietal fMRI activation in their sample, with the TMS effect only observed when the right hemisphere was stimulated, isn't the first and maybe most interesting aspect in the current study that already at this level, the replication shows a much more nuanced picture? Maybe there is no replicable or robust right IPS dominance here, OR there is but only at the novice level of task execution , and with practise this right IPS activity disappears (is shifted to other regions, less needed, etc). This would be in line with similar research on the test retest reliability of BOLD fMRI). And this would elegantly explain the lack of TMS effects when still sticking to an a priori right IPS hotspot for individual TMS targeting even in the absence of an individual fMRI activation at the moment of the TMS.

In his context I really enjoyed this extra analyses " Further, we performed an additional analysis comparing activation in the first two fMRI blocks recorded during the task with the second two blocks, and demonstrated there was a dramatic drop in functional activation from an initially significant right IPS activation. This decrease could have resulted from practice, learning, and/or fatigue, and led to a non-significant activation when combining all 4 blocks. As this provides some insight into targeting issues that may have hindered us from finding an rTMS effect, we have included this additional analysis in the manuscript"

For me this deserves more attention as this indeed relates to the heart of the ultimate goal of individualised targeting. If an fMRI activation changes dynamically with practise (as is the case here), how do I select my individual hotspot for TMS targeting then? I think just following what previous studies have found even though the fMRI data is different (for what reason ever, practise, age, pathology) may not be the best choice. Please note that Cohen Kadosh and sack cannot count as 2 independent validations of right IPS activation using fMRI / TMS as these studies were not independent from each other.

This is what I meant when I suggested to rather target the individual fMRI hotspots of this task in this sample and at this level of practise, even if it differs from the right IPS.

Best wishes,

Alex

Reviewer #2: The authors did a good job responding to my and other Reviewer's commentary. This represents an important contribution to our developing field.

Reviewer #4: The authors have addressed my comments, and have revised the manuscript so that it is now a direct reflection of what the results show, and can impart to the field as a result.

7. PLOS authors have the option to publish the peer review history of their article (what does this mean?). If published, this will include your full peer review and any attached files.

Reviewer #1: **Yes: **Alexander T. Sack

Reviewer #2: No

Reviewer #4: No

---

## [Editor Report · Acceptance letter]

24 Apr 2024

PONE-D-23-36866R1 

PLOS ONE

Dear Dr. Beynel, 

I'm pleased to inform you that your manuscript has been deemed suitable for publication in PLOS ONE. Congratulations! Your manuscript is now being handed over to our production team.

Kind regards, 

on behalf of

Dr. Desmond J. Oathes 

Academic Editor

PLOS ONE